# A novel autotransfusion device saving erythrocytes and platelets used in a 72 h survival swine model of surgically induced controlled blood loss

Kévin Schreiber[1], Benoit Decouture[2], Audrey Lafragette[1], Stéphane Chollet[2‡], Marine Bruneau[2‡], Maxence Nicollet[2‡], Catherine Wittmann[2‡], Francis Gadrat[2‡], Alexandre Mansour[3‡], Patricia Forest-Villegas[2‡], Olivier Gauthier[1,4], Gwenola Touzot-Jourde[1,4] *

1 CRIP, Center for research and preclinical investigation, Oniris Nantes Atlantic College of Veterinary Medicine, Food Science and Engineering, Nantes, France, 2 i-SEP, Nantes, France, 3 CHU Rennes, Department of Anesthesiology Critical Care Medicine and Perioperative Medicine, Inserm CIC 1414 (Centre d'Investigation Clinique de Rennes), Université de Rennes, Rennes, France, 4 INSERM, UMRS 1229 RMeS (Regenerative Medecine and Skeleton), University of Nantes, ONIRIS, Nantes, France

☯ These authors contributed equally to this work.
‡ SC, MB, MN, CW, FG, AM, and PFV also contributed equally to this work.
* gwenola.touzot-jourde@oniris-nantes.fr

**Data Availability Statement:** All relevant data are within the manuscript and its Supporting

## Abstract

### Background

The purpose of this study was to develop a swine model of surgically induced blood loss to evaluate the performances of a new autotransfusion system allowing red blood cells and platelets preservation while collecting, washing and concentrating hemorrhagic blood intraoperatively.

### Methods

Two types of surgically induced blood loss were used in 12 minipigs to assess system performance and potential animal complications following autotransfusion: a cardiac model (cardiopulmonary bypass) and a visceral model (induced splenic bleeding). Animal clinical and hematological parameters were evaluated at different time-points from before bleeding to the end of a 72-hour post-transfusion period and followed by a post-mortem examination. System performances were evaluated by qualitative and quantitative parameters.

### Results

All animals that received the autotransfusion survived. Minimal variations were seen on the red blood cell count, hemoglobin, hematocrit at the different sampling times. Coagulation tests failed to show any hypo or hypercoagulable state. Gross and histologic examination didn't reveal any thrombotic lesions. Performance parameters exceeded set objectives in both models: heparin clearance ($\geq$ 90%), final heparin concentration ($\leq$ 0.5 IU/mL), free

Information files. Additional data may be ask directly to the corresponding author.

**Funding:** The study was funded by BPi France (French Public Bank for investments) by the grant PSPC-402492 (BPI grants for structuring competitive research and development). The grant was awarded to the consortium Isep/Oniris/ Université de Rennes. The funders had no role in study design, data collection and analysis, decision to publish, or preparation of the manuscript.

**Competing interests:** Dr. Benoit Decouture (BD) is currently employed as project manager by i-SEP (Nantes, France). Stéphane Chollet (SC) is currently employed as technical director by i-SEP (Nantes, France). Marine Bruneau (MB) was employed as a research and development engineer by i-SEP (Nantes, France). Maxence Nicollet (MN) was employed as a technician by i-SEP (Nantes, France). Dr. Catherine Wittman (CW) was employed as a compliance and clinical affairs deputy head by i-SEP (Nantes, France). Dr. Francis Gadrat (FG) is currently employed as medical director by i-SEP (Nantes, France). Dr. Patricia Forest-Villegas (PVF) is currently employed as scientific director by i-SEP (Nantes, France). The authors confirm the fact that some of the co-authors are employed by i-SEP does not alter their adherence to PLOS ONE policies on sharing data and materials. All data generated during the study are fully available and all relevant data have been incorporated into the manuscript.

hemoglobin washout ($\geq 90\%$) and hematocrit (between 45% and 65%). The device treatment rate of diluted blood was over 80 mL/min.

## Conclusions

In the present study, both animal models succeeded in reproducing clinical conditions of perioperative cardiac and non-cardiac blood loss. Sufficient blood was collected to allow evaluation of autotransfusion effects on animals and to demonstrate the system performance by evaluating its capacity to collect, wash and concentrate red blood cells and platelets. Reinfusion of the treated blood, containing not only concentrated red blood cells but also platelets, did not lead to any postoperative adverse nor thrombogenic events. Clinical and comparative studies need to be conducted to confirm the clinical benefit of platelet reinfusion.

## Introduction

Allogenic blood transfusions have a global acceptable safety profile in countries with well-developed blood banking systems, but are associated with the transmission of infectious disease, transfusion reactions, and immunosuppression [1–4]. Limited availability of donors and high processing costs place further constraints on the use of allogenic transfusions [5–8]. Consequently, safe and cost-efficient alternatives to allogenic transfusions are highly requested by health care providers and anesthesiologists for the patient's benefit [9, 10].

Autotransfusion devices allow collection of shed blood during a surgical procedure and represent an attractive method for reducing allogenic blood transfusion. Benefits of auto-transfusion have been demonstrated in multiple surgical contexts [11, 12]. Autotransfusion devices play an important role in patient blood management and are recommended by international guidelines [13, 14]. More precisely, the use of autotransfusion devices has been shown to reduce the need for allogenic blood transfusions in surgical procedures (cardiac, orthopedic, visceral, vascular) by 38% (RR 0.62; 95% CI 0.55–0.70) [11]. Once a minimal blood volume of 400–600 mL has been collected from the surgical site, it can be filtered, washed, concentrated, and re-transfused to the patient [15]. Currently available cell salvage devices process the collected blood mainly by centrifugation and produce packed red blood cells (RBCs) with a satisfactory safety profile for the average patient. However, centrifugation-based processing has been questioned due to damaged blood cells contributing to adverse outcomes in critical patients such as immunocompromised or neonatal patients [16]. Moreover, the process usually removes platelets and coagulation factors when used with standard programs intraoperatively and, when treating large volumes, can result in a dilutional coagulopathy [17–19].

Platelets are critical regulators of hemostasis, serving as the matrix for initial vascular plug formation, creating a scaffold for the generation of fibrin clots, and releasing pro-wound-healing cytokines and procoagulant microparticles [20, 21]. Platelet concentrate reinfusion has been largely used in order to prevent or to treat perioperative hemorrhage in thrombocytopenic patients but also in combination with red blood cells and therapeutic plasmas for cases of massive bleeding due to trauma and surgical or obstetrical procedures [22, 23]. Many studies have been conducted to evaluate efficient treatments to quickly reach hemostasis and hence improve patient recovery and survival chances. In the past ten years,

studies showed a regained interest for whole blood transfusion, specifically in cases of hemorrhagic syndromes [24]. As such, results obtained from the military field demonstrate the advantages of bringing as early as possible all factors, including platelets in combination with RBCs [25, 26]. Current recommendations state that platelet transfusion should be performed as soon as possible, even if it needs to be repeated in case of persistent bleeding, and in association with RBC concentrate transfusion as soon as necessary [27]. The need for efficient blood salvage processing and autotransfusion of concentrates comprised of both RBCs and platelets has been identified as well as the necessity to improve current technology and develop new technology [28, 29].

The novel autotransfusion device evaluated in this study is a medical device intended to collect and wash intra-operatively collected blood to obtain a concentrated blood product at the end of the process. The device integrates an innovative filtration technology through hollow fibers, allowing RBCs concentration but also platelets within the concentrated blood product and contaminant elimination. A preliminary *in vitro* study demonstrated the ability to efficiently wash and concentrate red blood cells, white blood cells and platelets without significant impact on cell integrity and function [30]. The dual objective of this study was first to develop a swine surgical model of controlled blood loss with a 72-hour post-operative follow-up, and then to evaluate the use of the autotransfusion device in this swine model. Device output, salvaged blood characteristics and a 72-hour post transfusion follow-up of the animals were used to assess suitability and performance.

## Materials and methods

### Autotransfusion device and study design

The autotransfusion system tested is a medical device (Same™, i-SEP, France) consisting of reusable equipment and disposable consumables. The reusable equipment is composed of a roller stand supporting a structure integrating the electrical, electronic, computer and mechanical elements allowing the device to treat the blood. Consumables consists in a suction and anticoagulation line, a blood collection reservoir and a treatment set. The suction line allows collection and anticoagulation of the blood obtained from the surgical site. A rough filtration before blood storage is performed in the blood collection reservoir. The treatment set consists mainly of tubings, a hollow fiber cartridge to separate blood cells from plasma, a blood treatment bag that ensures blood washing, a waste bag collecting plasma and contaminants, and a reinfusion bag to store filtered, washed and concentrated blood. A more detailed processing is described elsewhere [30].

The study was designed to develop two surgical models of controlled blood loss (abdominal and cardiac). The abdominal visceral model consisted of a splenic bleeding by surgically-induced capsule injuries representative of non-cardiac bleeding with aspiration of shed blood partially coagulated. Conversely, during the cardiac procedure, the blood was highly anticoagulated and collected from the right lateral thoracotomy surgical site and from the cardio-pulmonary bypass venous reservoir after ending an hour-long extracorporeal circulation. All collected blood were treated with the device to test the performance and evaluate a 72 h animal survival and post-mortem findings after autotransfusion. Taking into account the 3R recommendations, it was estimated that 4 to 6 animals per surgical model would be necessary to account for individual animal variation as well as surgical technique refinement and animal management optimization. Due to the early stage of the device testing on animal recovery, no randomization was used and the animal subjects were allocated to the cardiac group depending on their thoracic conformation that would facilitate the cardiac surgical approach by right lateral thoracotomy.

## Animals

The animal study protocol received approval from the regional ethical committee for animal experimentation and was authorized by the French Ministry of Higher Education, Research and Innovation (Apafis number #11079-2017071317574824v1). Twelve adult female Yucatan mini-pigs were included in the study. The animal management, anesthesia and analgesia protocol as well as perioperative care, euthanasia and post-mortem examination are described in details in the S1 Appendix in S1 File. Briefly, after a week of acclimatization in an enriched environment, the animals were anesthetized by intramuscular injection for instrumentation, surgically induced controlled bleeding and autotransfusion. During anesthesia maintained with a balanced technique, controlled ventilation was instituted and jugular access was gained to place a central line for repeated blood sampling. Continuous monitoring during anesthesia included ECG, blood pressure measurement indirectly during instrumentation and invasively during the surgically induced bleeding and the following transfusion, pulse oximetry, inspired and expired gas analysis, arterial blood gases and body temperature. Anesthesia depth and intravenous fluid replacement were adjusted to the animal response to acute blood loss. Postoperative care comprising analgesia and supportive therapy was adapted to surgical models and each individual recovery during a 72 h follow-up post-transfusion. At the end of the period, general anesthesia was reinduced before euthanasia. Post-mortem examination was geared to assess thrombogenic risk according to the ISO standard 10933–4 [31] and the application of the FDA guide: Use of International Standard ISO 10993–1, "Biological evaluation of medical devices—Part 1: Evaluation and testing within a risk management process", section thrombogenicity, published in June 2016 and updated in 2021 [32].

## Surgical models of controlled blood loss

**Abdominal visceral model of controlled blood loss.** A 20 cm long midline laparotomy incision was performed to exteriorize the spleen from the abdominal cavity. To counteract known porcine hypercoagulability [33–35] and to facilitate blood loss and collection, a low dose of heparin (25 IU/kg IV) was administered just before initiating the splenic injury and repeated if needed during bleeding. This dosage has been used in previous studies of porcine intra-abdominal hemorrhage and has shown to bring the porcine coagulation profile within values found in humans [36]. ACT measurements (Activated Clotting Time, via Medtronic ACT II Coagulation Timer) were done before heparin administration, repeated during the blood collection time (target ACT 90–130 s) and just before transfusion to ensure a return to baseline ACT value.

Multiple lesions were created into the splenic parenchyma using digitoclastic technique and an additional lesion was done by severing the splenic vein. Spontaneous bleeding was let to drip into the abdominal cavity and then aspirated through the suction line (depression kept under 200 mbar to minimize hemolysis) [37] into the autotransfusion device blood collection reservoir. After reaching a 900 to 1100 ml volume of a blood and anticoagulation solution mixture in the collection reservoir, corresponding to a minimal of 20% of the estimated animal blood volume (61–68 ml/kg) [38], the bleeding was controlled by removal of the spleen using a tissue fusion device (Atlas Ligasure™ forceps 10 mm, Covidien, Medtronic). The surgical wound was sutured in different layers. A protective bandage consisting of an adhesive padded tape (Animal Polster, Snogg, Norway) was placed to cover the incision.

**Cardiac model of controlled blood loss.** A right lateral thoracotomy was performed in the third intercostal space. Cardio-pulmonary bypass (CPB) was set in place by cannulation of the right atrium and the ascending aorta. Extra-corporeal circulation (priming volume of 1250 mL of Ringer Lactate solution) was then initiated and maintained for an hour. Classic

anticoagulation with heparin (5000 IU in the priming volume, 300 IU/kg IV to the animal) and ACT monitoring (> 400 s) were used. At the end of the bypass, adequate animal volemia was restored based on hemodynamic monitoring data. CBP cannulas were removed and cannulated vessels sutured. Heparin effect reversal was achieved with a slow IV infusion of protamine (1 mg/100 IU of heparin used during surgery). To allow elimination of residual air and fluids from the thoracic cavity, a chest tube was placed at the end of the procedure and connected to a low negative pressure drainage system. The surgical wound was closed in layers. A wound catheter was left in place to allow instillation of local anesthetics. A protective bandage consisting of an adhesive padded tape (Animal Poltser, Snogg, Norway) was placed to cover the incision. Hemorrhagic blood aspirated from the chest cavity (suction pressure kept under 200 mbar to minimize hemolysis) and blood remaining in the CPB circuit after stopping the extra-corporeal circulation were collected to be treated by the autotransfusion device (minimal volume of 700 mL).

## Laboratory testing

Blood samples were collected from before surgery to the end of the 72-hour follow up to monitor possible variations in hematologic and coagulation parameters (Fig 1).

Times of sampling were before surgery once the central venous catheter was in place (T0), at the end of the transfusion (TP), then postoperatively between 2 h and 6 h (T2-6h), 6 h and 12 h (T6-12h), then every 12 hours until euthanasia at 72 h post-transfusion (T12-24h, T24-36h, T36-48h, T48-60h, T60-72h). Total blood volume collected was refined by establishing the minimal volume needed for laboratory testing ahead of the experiment. All blood samplings were performed by gentle aspiration through the central venous catheter which allowed stress-free repetitive interventions on the minipigs post-operatively. Additional blood samples were obtained from the blood collection reservoir (TR) and from the transfusion bag (TB) before reinfusion started. Blood analyses consisted of complete blood count (CBC) including RBC, WBC and platelet count, hematocrit and total hemoglobin (Procyte Dx Hematology

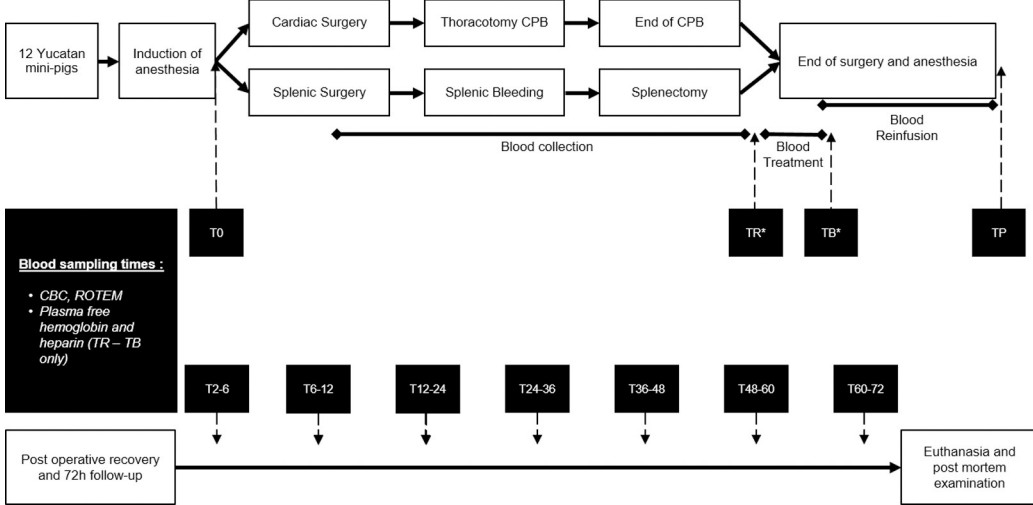

**Fig 1. Schematic description of the study design.** The different times of sampling are during anesthesia before blood loss (T0), in the blood collection reservoir before the treatment (TR*), in the blood bag after the treatment (TB*), after autologous transfusion at the end of anesthesia (TP), then during the 72 h survival follow-up period (7 sampling times following the end of the transfusion: T2-6h, T6-12h, T12-24h, T24-36h, T36-48h, T48-60h and T60-72h). CPB: Cardio Pulmonary Bypass; CBC: Complete blood count. *TR and TB samplings were taken from the blood collection reservoir and the transfusion blood bag and not sampled directly in the animal bloodstream.

Analyzer, IDEXX, Hoofddorp, The Netherlands), plasma free hemoglobin concentration measurement for the calculation of the hemolysis rate (Plasma/Low Hb photometer, HemoCue AB, Ängelholm, Sweden), plasma heparin concentration measurement through anti-Xa activity assay (Heparin standard-HNF, HemosIL Liquid anti-Xa, ACLTOP, IL, Werfen, Le Pré-Saint-Gervais, France) and ROTEM tests to evaluate coagulation status and platelet aggregation (ROTEM delta and ROTEM platelet, Werfen, Le Pré-Saint-Gervais, France).

Analyses of the coagulation status with ROTEM tests included EXTEM, INTEM and HEP-TEM tests. HEPTEM is the same test as the INTEM but with additional heparinase. Both tests were performed to take into account the fact that heparin was administered to all animals during surgery and was used for the suction line anticoagulation. Analyzed parameters were clotting time (CT), amplitude at 30 minutes (A30) and maximum clotting firmness (MCF). Analyses of the platelet aggregations were performed with the ROTEM platelet module. Platelet activation was induced by the ADP agonist provided by Werfen.

## Autotransfusion device evaluation

Qualitative and quantitative parameters were used to evaluate the performance of the autotransfusion system in washing and concentrating collected blood. Quality of the treated blood was assessed by measuring heparin and free hemoglobin concentrations and calculating their clearance from the washed blood. As defined by the American Association Blood Bank (AABB) [39], the threshold concentration of residual heparin in the concentrated blood is $\leq 0.5$ IU/mL and the washout is expected to be $\geq 90\%$. Regarding hemolysis, the AABB states that the residual free hemoglobin concentrations may be quite high (5–10 g/L) despite greater than 90% removal (AABB recommendations). Washout, expressed as component clearance (%), was calculated with the following formula:

$$\text{Heparin clearance } (\%) = \frac{(\text{Initial quantity of Heparin} - \text{Final quantity of Heparin})}{\text{Initial quantity of Heparin}} \text{ x } 100$$

Regarding free hemoglobin, a formula was adapted taking into account the free hemoglobin production during the treatment. The calculation is based on the eliminated free hemoglobin in the waste:

$$\text{Free hemoglobin washout } (\%) = \frac{(\text{Free hb in the waste})}{(\text{Free hb in the waste} + \text{Free hb in the concentrated blood})} \text{ x } 100$$

Eliminated free hemoglobin (free hemoglobin in the waste bag) is calculated by the difference between the total amount of hemoglobin measured in the blood after the treatment compared to the total amount of hemoglobin measured in the blood before the treatment. Hemolysis rate is the percentage of the hemoglobin amount present in the extracellular medium (free hemoglobin) compared to the total amount of hemoglobin in the whole blood. The extracellular medium volume is calculated by subtracting the hematocrit percentage from the whole blood volume (100%). The formula to calculate hemolysis with the appropriate correction is therefore [40]:

$$\text{Hemolysis rate} = \frac{(100 - \text{Hematocrit}) \text{ x Free hemoglobin}}{\text{Total hemoglobin}}$$

Quantitative performance parameters used to describe the treated blood product included cell yields (RBCs, Platelets and WBCs), the final hematocrit and the final hemoglobin concentration of the blood product ready to be transfused. Cell yields are calculated with the following

formula:

$$\text{Cell yield (\%)} = \left( \frac{\text{Initial volume containing cells x Initial concentration of cells}}{\text{Final volume containing cells x Final concentration of cells}} \right) \text{x } 100$$

Duration of a treatment cycle was used to characterize the technical efficacy of the device. Expected device performance was to obtain a heparin clearance $\geq$ 90% and a residual heparin concentration $\leq$ 0.5 IU/mL, a free hemoglobin washout $\geq$ 90%, a RBC yield $\geq$ 80%, a final hematocrit between 45% and 65% and a treatment rate $\geq$ 80 mL/min.

### Statistical analysis

Due to the observational nature of the study on small animal samples, the results are reported by descriptive statistics, expressed as individual value or as median (min-max) for each surgical model. Data from the visceral model are represented with circles while data from the cardiac model are represented as squares. Each series of points represent a unique animal on figures.

## Results

### Data on animals

Twelve one year-old female Yucatan mini-pigs were included in the study with a median (min-max) weight of 46.9 (45.75–48.25) kg in the visceral model (n = 5) and 48 (41–53) kg in the cardiac model (n = 7). Each animal was individually identified by an ear tag (since a few days after birth) to guarantee traceability.

Three animals in the cardiac model were excluded due to premature death before administration of the collected and treated blood. Two minipigs died despite resuscitation attempts from ventricular fibrillation during the thoracotomy, one before the bypass started and one at the time of aortic cannula removal, and the third mini-pig had a fatal anaphylactic reaction during slow protamine infusion. All animals that underwent the complete surgical procedure and autotransfusion survived and were euthanized at the end of a 69 (68–71) h follow-up period after transfusion completion. Recovery from the visceral procedure was uneventful, mini-pigs stood up within the first two hours following the end of anesthesia, had all resumed eating and drinking at the time of the 6h post-transfusion clinical examination. No rescue analgesia was needed. In the cardiac model, nasal oxygen supplementation was needed for the first 8 to 18 hours of recovery and opioid analgesia could be stopped after the 24 to 36 hours of recovery. Mini-pigs stood up between 14 and 20 h post-transfusion and resumed eating shortly thereafter (0 to 4 h from standing).

Post-mortem examination in the visceral model showed gross and histological abnormalities that included a slight fibrino-congestive diffuse peritonitis in one animal and a slight focal perihepatitis in two animals, a hemorrhagic mediastinal lymph node in one animal, a renal infarct scar in two animals. Signs of inflammation and hemorrhage were considered to be results of the surgical trauma and the renal lesion was of chronic nature dating the injury well prior the surgery. In the cardiac model, lesions at the surgical site were identified in all animals: localized myocarditis, endocarditis and pericarditis at level of the right atrium, pericarditis in the left ventricle and right-sided pleuritis. One cardiac minipig had a multifocal granulomatous pneumonia associated with identification of inhaled vegetal particles. Gross and histologic examination failed in both models to reveal any thrombotic lesions consecutive to the autotransfusion.

### Laboratory testing

The complete blood count and ROTEM results are detailed in the (S1 and S2 Tables in S1 File respectively).

**RBC counts, hemoglobin and hematocrit.** In the visceral model, red blood cell (RBC) count, hemoglobin and hematocrit tended to be lower at the T0 sample (during anesthesia and before surgery) compared to the TP sample taken at the end of transfusion (RBC count: 4.6 [4.3–6.1] $10^6$/μL for T0 vs 5.4 [5.3–5.8] $10^6$/μL for TP; Hemoglobin in g/L: 93 [85–119] for T0 vs 111 [105–114] for TP; Hematocrit in %: 29 [27–39] for T0 vs 35 [34–38] for TP). During follow-up, no major variation of the RBC counts, the hemoglobin concentration or the hematocrit was noticed (Fig 2, panel A, B and C and S1 Table in S1 File).

In the cardiac model, minimal variations were seen on the RBC count, hemoglobin and hematocrit between T0 and TP samples (RBC count: 4.6 [4.3–5.1] $10^6$/μL at T0 vs 4.3 [4.1–4.5] $10^6$/μL at TP; Hemoglobin in g/L: 94.5 [90–105] at T0 vs 88.5 [85–93] at TP; Hematocrit in %: 30 [29–33] at T0 vs 28 [26–30] at TP) (Fig 3, panel A, B and C and S1 Table in S1 File). During follow-up, no major variation of red blood cell count, hemoglobin concentration nor hematocrit was noticed.

**WBC counts.** All the minipigs in both surgical models developed a neutrophilic leukocytosis while under anesthesia or in the early phase of recovery with a peak between 2 and 12 hours after the end of transfusion lasting for 24 hours (22.7 [15.8–27.9] $10^6$/μL at T2-6h and 28.2 [19.6–28.7] $10^6$/μL at T6-12h for the visceral model and 21.5 [16.0–29.5] $10^6$/μL at T2-6 and 26.3 [16.8–35.0] at T6-12h for the cardiac model) (Fig 4A and 4B; complete WBC values in S1 File).

**Platelet counts.** In both models, platelet counts decreased in post-operative blood samples. The nadir was reached between 6 and 12 hours post-operatively in both models. Thereafter, platelet count increased until a return close to normal count at 72 h post-operatively (see Fig 5 and S1 Table in S1 File)

**Coagulation control in the visceral model.** In the visceral model, baseline ACT were comprised between 69 and 96 s (median of 73 s) and increased between 102 and 133 s (median of 109 s) at 15 minutes post-heparin injection.

**Coagulation evaluation by ROTEM.** EXTEM and HEPTEM results are displayed in Tables 1 and 2 while INTEM results are to be found in S2 Table in S1 File. In the visceral model, EXTEM results showed a stable clotting time during the follow-up. In the cardiac model, EXTEM clotting time was more variable and tended to be slightly increased during the follow-up compared to before the surgery (T0). No clotting time exceeded 100 s. The HEPTEM results showed more variability in the clotting time that tended to increase slightly during the follow-up compared to before the surgery (T0; Table 2). The other parameters (A30, MCF and Alpha angle) showed minimal variations.

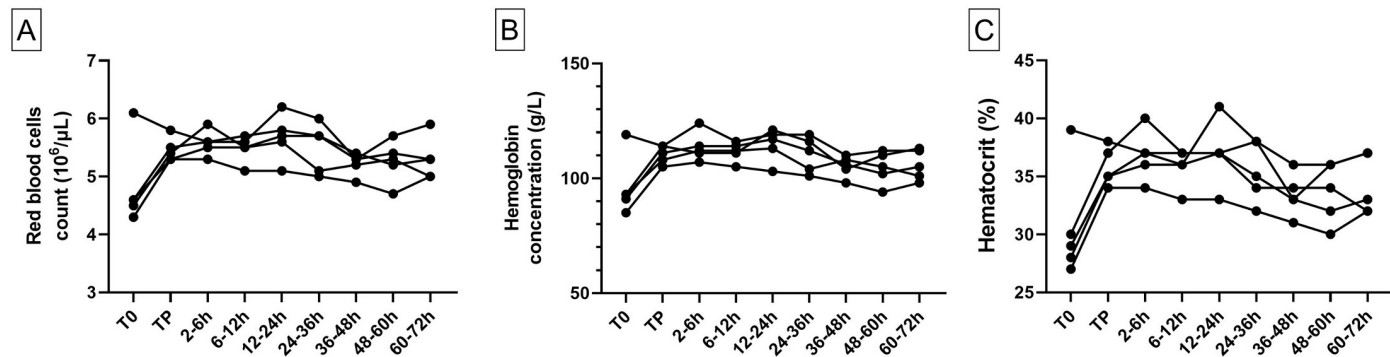

**Fig 2. RBC count, hemoglobin and hematocrit animal values in the visceral model (n = 5).** RBC count (A), hemoglobin (B) and hematocrit (C) are represented for each individual animal. The different times of sampling are during anesthesia before blood loss (T0) and after autologous transfusion at the end of anesthesia (TP), then during the 72 h survival follow-up period (7 sampling times following the end of the transfusion).

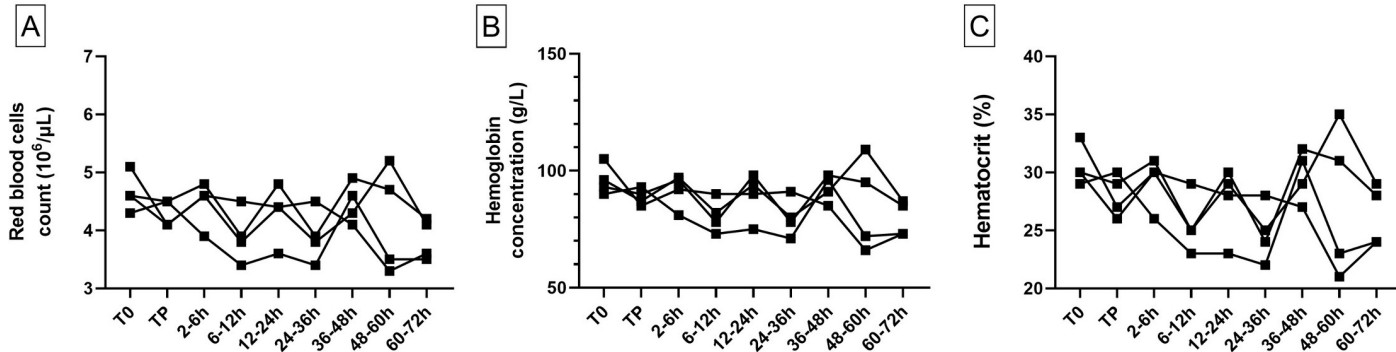

**Fig 3. RBC count, hemoglobin and hematocrit animal values in the cardiac model (n = 4).** RBC count (A), hemoglobin (B) and hematocrit (C) are represented for each individual animal. The different times of sampling are during anesthesia before blood loss (T0), after autologous transfusion at the end of anesthesia (TP), then during the 72 h survival follow-up period (7 sampling times following the end of the transfusion).

**ADP-induced platelet aggregation.** Whole blood platelet aggregation was induced by ADP and measured with the ROTEM platelet module. Results showed that AUC related to ADP-induced platelet aggregation was not modified at the post-operative sample time (TP) or during the follow-up (from 2–6 h to 60–72 h) compared to the pre-operative value (T0), in both visceral and cardiac models (Table 3).

## Controlled blood loss and autotransfusion device evaluation

The splenic surgical injury resulted in an active hemorrhage and a blood collection in a median time of 31 (22–51) min. End volumes in the collection reservoir ranged between 925 and 1100 mL (median 1000 mL) in the visceral model corresponding to an estimated blood loss of 600 to 900 mL representing a minimum of 20% of the animal blood volume. Volume in the collection reservoir after the end of the bypass was comprised between 900 and 1425 mL (median 1125 mL) in the cardiac model. The hematocrit of the collected blood in the reservoir (TR) for the visceral and the cardiac model was respectively 24 (16–28) % and 19 (16–21) %, with a hemolysis rate of 2.1 (0.6–2.4) % and 0.2 (0.1–0.5) %.

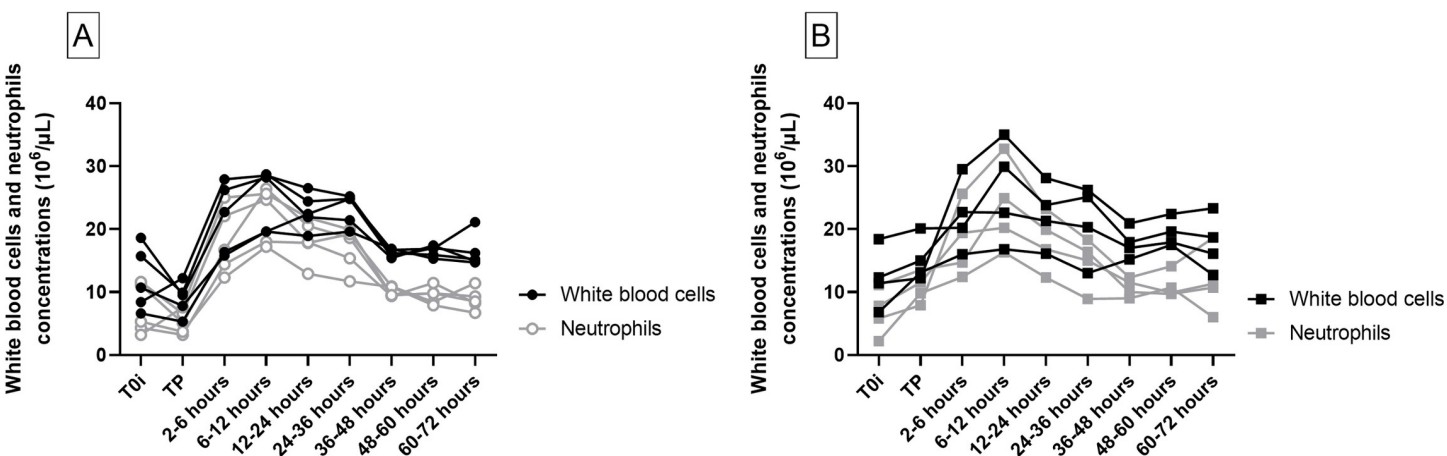

**Fig 4. WBC and neutrophil counts in the visceral model (A, n = 5) and the cardiac model (B, n = 4).** Each line represents one animal. The different times of sampling are during anesthesia before blood loss (T0), after autologous transfusion at the end of anesthesia (TP), then during the 72 h survival follow-up period (7 sampling times following the end of the transfusion).

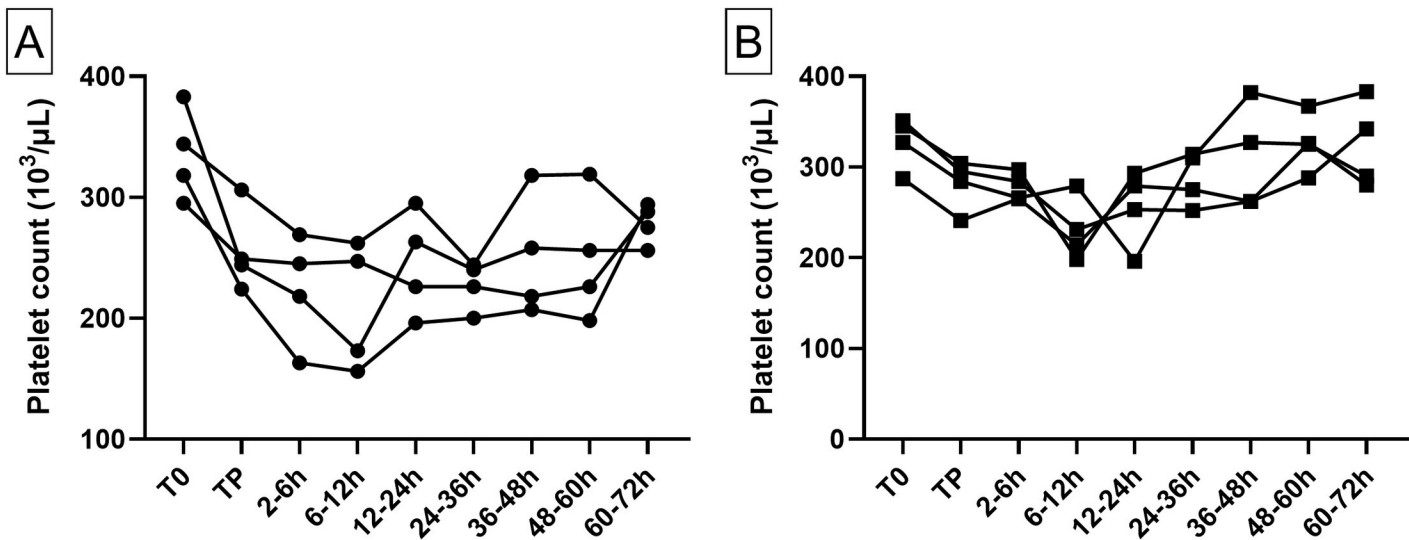

**Fig 5. Animal platelet counts in the visceral model (A, n = 4) and the cardiac model (B, n = 4).** Each line represents one animal. The different times of sampling are during anesthesia before blood loss (T0), after autologous transfusion at the end of anesthesia (TP), then during the 72 h survival follow-up period (7 sampling times following the end of the transfusion). One animal in the visceral model was excluded for the platelet count due to platelet aggregation verified on the blood smear and failure of the hematological analyzer.

The autotransfusion system performed a complete treatment cycle of 670 (670–755) mL in 5 min 54 s (5 min 27 s– 6 min 05 s) for the visceral group and 708 (670–755) mL in 6 min 10 s (4 min 0 s– 6 min 25 s) for the cardiac group. These treatment timings correspond to treatment rates at 121 (110–139) mL/min for the visceral group and 114 (106–189) mL/min for the cardiac group. Transfusion bags contained 229.4 (163.2–311.7) g of RBC solubilized in isotonic sodium chloride with a median hematocrit of 50 (45–60) %. Treated blood was transfused back to the animals in 23 (11–36) min in the visceral model and 29 (25–29) min in the cardiac model. The blood treatment was evaluated according to qualitative and quantitative parameters (Table 4). The quality of the reinfused blood was evaluated by measuring the concentration of unwanted substances, in particular heparin. Results show that the heparin concentration was always below 0.5 IU/mL except for one cardiac reinfused blood which was

**Table 1. ROTEM-EXTEM parameters during anesthesia and the 72 h survival follow-up period (9 sampling times).**

| | Parameters/Time | Surgical Models | T0 | TP | T2-6h | T6-12h | T12-24h | T24-36h | T36-48h | T48-60h | T60-72h |
|---|---|---|---|---|---|---|---|---|---|---|---|
| **EXTEM** | **CT (s)** | VISC | **56** *(49–65)* | **59** *(56–72)* | **62** *(44–77)* | **62** *(46–67)* | **65** *(55–78)* | **62** *(53–89)* | **75** *(51–83)* | **79** *(52–89)* | **67** *(54–71)* |
| | | CARD | **56.5** *(53–61)* | **51** *(44–61)* | **81.5** *(55–97)* | **70.5** *(67–83)* | **61** *(40–85)* | **71.5** *(68–77)* | **62.5** *(50–87)* | **70** *(68–74)* | **85** *(78–88)* |
| | **A30 (mm)** | VISC | **73** *(70–76)* | **67** *(61–69)* | **69** *(63–69)* | **69** *(63–70)* | **72** *(71–74)* | **75** *(70–75)* | **75** *(63–76)* | **76** *(73–79)* | **78** *(72–80)* |
| | | CARD | **71.5** *(67–81)* | **72** *(68–75)* | **71** *(68–73)* | **71.5** *(67–73)* | **74** *(72–77)* | **77** *(74–79)* | **76.5** *(74–81)* | **79.5** *(74–83)* | **79.5** *(61– 83)* |
| | **MCF (mm)** | VISC | **74** *(71–76)* | **68** *(62–73)* | **70** *(64–72)* | **71** *(64–71)* | **75** *(74–76)* | **77** *(73–78)* | **78** *(66–79)* | **79** *(76–81)* | **80** *(75–82)* |
| | | CARD | **73** *(69–82)* | **72.5** *(68–76)* | **71.5** *(70–74)* | **73** *(68–74)* | **76** *(74–78)* | **78.5** *(76–81)* | **78** *(77–82)* | **81** *(76–84)* | **80.5** *(63–85)* |
| | **Alpha Angle (°)** | VISC | **79** *(78–81)* | **77** *(74–80)* | **77** *(76–78)* | **76** *(72–77)* | **77** *(76–78)* | **78** *(74–79)* | **78** *(75–78)* | **78** *(77–81)* | **80** *(78–80)* |
| | | CARD | **78** *(75–81)* | **76.5** *(74–80)* | **76.5** *(75–79)* | **77** *(74–78)* | **77** *(76–79)* | **77** *(77–78)* | **77** *(75–78)* | **77** *(76–81)* | **78** *(67–80)* |

Results expressed as median (min-max) for the visceral (VISC, n = 5) and cardiac models (CARD, n = 4)

The different times of sampling are during anesthesia before blood loss (T0), after autologous transfusion at the end of anesthesia (TP), then during the 72 h survival follow-up period (7 sampling times following the end of the transfusion). CT: clotting time, A30: amplitude at 30 minutes, MCF: maximum clotting firmness.

**Table 2. ROTEM-HEPTEM parameters during anesthesia and the 72 h survival follow-up period (9 sampling times).**

| Parameters/ Time | | Surgical Models | T0 | TP | T2-6h | T6-12h | T12-24h | T24-36h | T36-48h | T48-60h | T60-72h |
|---|---|---|---|---|---|---|---|---|---|---|---|
| HEPTEM | CT (s) | VISC | **147** *(141–157)* | **154** *(146–178)* | **132** *(114–161)* | **141** *(104–162)* | **158** *(132–167)* | **168** *(154–201)* | **181** *(133–222)* | **177** *(171–218)* | **164** *(118–187)* |
| | | CARD | **189** *(119–218)* | **170** *(149–245)* | **171.5** *(153–188)* | **146** *(106–192)* | **148.5** *(133–182)* | **148.5** *(138–199)* | **171** *(131–190)* | **187.5** *(157–199)* | **195** *(144–217)* |
| | A30 (mm) | VISC | **66** *(65–67)* | **62** *(58–63)* | **60** *(58–63)* | **62** *(58–64)* | **64** *(63–65)* | **66.5** *(65–68)* | **60** *(48–65)* | **67** *(64–71)* | **67** *(66–72)* |
| | | CARD | **62.5** *(46–68)* | **62.5** *(61–68)* | **65.5** *(63–67)* | **66** *(64–67)* | **67** *(45–68)* | **70** *(70–70)* | **72.5** *(69–79)* | **72.5** *(69–73)* | **73** *(67–74)* |
| | MCF (mm) | VISC | **69** *(67–71)* | **64** *(60–66)* | **65** *(60–70)* | **65** *(63–70)* | **70** *(68–71)* | **72.5** *(71–74)* | **70** *(55–73)* | **72** *(70–75)* | **75** *(72–77)* |
| | | CARD | **66.5** *(48–71)* | **66** *(63–70)* | **68.5** *(66–69)* | **69.5** *(68–71)* | **72** *(49–72)* | **75** *(74–75)* | **77** *(74–82)* | **76.5** *(74–77)* | **78** *(73–79)* |
| | Alpha Angle (°) | VISC | **81** *(78–81)* | **78** *(77–80)* | **78** *(76–80)* | **77** *(76–80)* | **78** *(77–79)* | **77.5** *(75–79)* | **75** *(65–78)* | **76** *(75–81)* | **79** *(74–81)* |
| | | CARD | **78.5** *(69–80)* | **78** *(76–81)* | **79** *(77–79)* | **77.5** *(76–79)* | **78** *(65–79)* | **79** *(76–79)* | **79** *(77–79)* | **78** *(78–81)* | **79** *(79–80)* |

Results expressed as median (min-max) for the visceral (VISC, n = 5) and cardiac models (CARD, n = 4)

The different times of sampling are during anesthesia before blood loss (T0), after autologous transfusion at the end of anesthesia (TP), then during the 72 h survival follow-up period (7 sampling times following the end of the transfusion).

CT: clotting time, A30: amplitude at 30 minutes, MCF: maximum clotting firmness

Additional ROTEM (INTEM) results are in the (S2 Table in S1 File).

at 0.6 IU/mL (0.14 [0.00–0.36] IU/mL for the reinfused blood from visceral model and 0.46 [0.44–0.60] IU/mL for the reinfused blood from the cardiac model). Heparin concentration in the blood before treatment was at 4.5 [3.4–6.8] IU/mL in the visceral model and 9.6 [8.5–11.5] IU/mL in the cardiac model. Qualitative and quantitative performance parameters (heparin clearance, free hemoglobin washout, hematocrit and hemoglobin concentration) exceeded set objectives, except for RBC yield and hemolysis rate (Table 4). Red blood cell yield was 61.7 (50.1–71.9) % for the visceral model and 70.5 (65.4–72.1) % for the cardiac model.

Platelet concentration was 54 (16–83) $10^3$/μL and 157 (115–192) $10^3$/μL in the collection reservoir (TR) and 109 (64–147) $10^3$/μL and 305 (258–337) $10^3$/μL in the transfusion bag, for the visceral and the cardiac model respectively. Platelet yield was 74.4 (40.3–150.7) % and 58.2 (49.1–62.2) % for the visceral and the cardiac model respectively (Table 4).

Coagulation properties of the blood from the collecting reservoir and in the transfusion bag were evaluated by ROTEM assays. EXTEM tests show that the clotting time was considerably increased in the transfusion bag compared to before filtration and washing for both models (for the visceral model, 56 [49–65] s before treatment vs 517 [291–3600] s after the treatment and for the cardiac model, 56.5 [53–61] s before the treatment vs 1769 [119–2277] s after).

**Table 3. Platelet aggregometry results.**

| Parameters/Time | | Surgical Models | T0 | TP | T2-6h | T6-12h | T12-24h | T24-36h | T36-48h | T48-60h | T60-72h |
|---|---|---|---|---|---|---|---|---|---|---|---|
| ROTEM Delta ADP-induced aggregation | Amplitude 6 minutes (ohm) | VISC | **24** *(10–32)* | **16** *(14–23)* | **20** *(12–30)* | **25** *(15–34)* | **22** *(10–34)* | **14** *(8–23)* | **19** *(12–29)* | **17** *(7–28)* | **23.5** *(11–28)* |
| | | CARD | **23** *(17–25)* | **15.5** *(8–32)* | **20.5** *(18–24)* | **19.5** *(18–27)* | **14.5** *(12–15)* | **12** *(7–15)* | **20** *(9–24)* | **12** *(5–18)* | **19** *(14–28)* |
| | AUC (A.U.) | VISC | **99** *(40–130)* | **72** *(60–94)* | **87** *(54–119)* | **100** *(64–136)* | **86** *(45–135)* | **89** *(45–97)* | **84** *(70–122)* | **98** *(55–121)* | **111.5** *(50–119)* |
| | | CARD | **94** *(77–106)* | **71.5** *(42–130)* | **88.5** *(80–105)* | **79.5** *(79–115)* | **60.5** *(55–64)* | **59.5** *(45–90)* | **96** *(73–102)* | **61** *(52–76)* | **121** *(82–122)* |

Aggregation amplitude at 6 min (ohm) and area under the curve in arbitrary unit (AUC, A.U.) are expressed as median (min-max) for the visceral (VISC, n = 5) and cardiac models (CARD, n = 4). The different times of sampling are during anesthesia before blood loss (T0), after autologous transfusion at the end of anesthesia (TP), then during the 72 h survival follow-up period (7 sampling times following the end of the transfusion).

**Table 4. Qualitative and quantitative assessment of the autotransfusion device.**

| Parameters | Visceral model N = 5 | Cardiac model N = 4 | Criteria |
|---|---|---|---|
| *Quantification of blood elements and quality of collected blood before treatment (TR)* | | | |
| Red blood cell concentration ($10^6$/µL) | **3.9** *(2.6–4.6)* | **3.0** *(2.5–3.3)* | **NA** |
| Hemoglobin concentration (g/L) | **80** *(52–92)* | **61** *(51–67)* | **NA** |
| Hematocrit (%) | **24** *(16–28)* | **19** *(16–21)* | **NA** |
| Hemolysis rate (%) | **2.1** *(0.6–2.4)* | **0.2** *(0.1–0.5)* | **NA** |
| Platelet concentration ($10^3$/µL) | **54** *(16–83)* | **157** *(115–192)* | **NA** |
| *Quality of the blood to be reinfused (TB)–unwanted substances* | | | |
| Anticoagulant (heparin) concentration (IU/mL) | **0.14** *(0.00–0.36)* | **0.46** *(0.44–0.60)* | **≤ 0.5** |
| Anticoagulant clearance (%) | **97.3** *(93.6–100.0)* | **98.3** *(95.6–99.0)* | **≥ 90%** |
| Hemolysis rate (%) | **1.2** *(1.0–2.0)* | **1.4** *(1.4–1.6)* | **NA** |
| Free hemoglobin washout (%) | **94** *(88–98)* | **90** *(87–94)* | **≥ 90%** |
| *Quantification of blood elements to transfuse (TB) and device performance* | | | |
| Red blood cell concentration ($10^6$/µL) | **7.5** *(7.0–8.9)* | **7.2** *(6.8–7.2)* | **NA** |
| Red Blood Cells Yield (%) | **61.7** *(50.1–71.9)* | **70.5** *(65.4–72.1)* | **≥ 80%** |
| Platelet concentration ($10^3$/µL) | **109** *(64–147)* | **305** *(258–337)* | **NA** |
| Platelet yield (%) | **74.4** *(40.3–150.7)* | **58.2** *(49.1–62.2)* | **≥ 40%** |
| Final Hematocrit (%) | **50** *(45–60)* | **49** *(45–50)* | **45% < value > 65%** |
| Total hemoglobin concentration (g/L) | **155** *(141–189)* | **152** *(139–159)* | **NA** |
| White Blood Cells yield (%) | **69** *(59–77)* | **83** *(72–88)* | **NA** |
| Treatment rate (mL/min) | **121** *(110–139)* | **114** *(106–189)* | **≥ 80 mL/min** |

Results expressed as median (min-max).

Such as on the animal blood, HEPTEM results show variability in the clotting time and tended to increase after treatment compared to before treatment (147 [141–157] s before vs 354 [274–3600] s after treatment in the visceral model and 189 [119–218] before vs 986 [174–2085] s after treatment in the cardiac model).

## Discussion

The aim of this study was to develop a minipig animal model of surgically induced hemorrhage reproducing the clinical conditions of perioperative blood loss and evaluate the performance of a new autotransfusion device allowing the concentration of not only red blood cells but also platelets using an innovative filtration technology. No clinical complication nor bleeding following the reinfusion and recovery from anesthesia were observed on the animals during the 72 h postoperative period. Device performance fulfilled the criteria that were defined either from regulatory limits, in particular for anticoagulant elimination and maximum acceptable hemolysis in the treated blood, or from scientific consensus in terms of performance on devices already available [39].

Several hemorrhagic models on different animal species have been used to evaluate the safety and performance of autotransfusion devices. Hofbauer et al. [41] described a canine model in which whole blood was collected on anticoagulated bags for further processing with an automated cell salvage device. Treated blood was not reinfused to the animals. Total recovery of RBCs was 80 ± 12%, but 70% of the platelets were washed out, while 57% of the leukocytes remained in the concentrated final product. Due to differences between canine and human blood [42] and for ethical reasons, this animal species was not considered as a potential model for the present study. Ovine and bovine models were not considered because ovine

RBCs are smaller than human RBCs and data published using fresh bovine blood did not include any evaluation of *in vivo* reinfusion [43–45].

In a porcine model, Vagianos et al. [46] evaluated hemorrhagic blood collection with citrated sponges instead of suction. Although the blood was not reinfused to the animals, they concluded that blood collection by means of surgical sponges may be a safe and an efficient method. Autotransfusion experiments in pigs were initially performed by reinfusion of a non-processed blood collected after a controlled hemorrhage that resulted in a 40% blood volume loss [47]. Animals were followed up for 48 hours without any mortality. The porcine model seems to be the most commonly used for blood transfusion studies [48–50]. The minipig model (*Sus Scrofa domesticus*) has been extensively used in hemorrhagic models and cardio-vascular research including testing medical devices [51, 52]. The Yucatan minipig has anatom-ical [53], physiological and blood characteristics comparable to those of humans, as well as a good homogeneity between the animals [54] which makes it a suitable model for *in vivo* bio-logical evaluation of an autotransfusion device.

To reproduce as close as possible to the clinical setting, two surgical models were developed: (i) a cardiac model consisting of a cardiopulmonary bypass after a lateral thoracotomy and treatment of the remaining blood in the CPB circuit and (ii) an abdominal visceral model of surgically induced splenic hemorrhage and *in situ* blood aspiration from the surgical site. The present study confirmed that both models were relevant to evaluate the safety and the perfor-mance of the autotransfusion device not only perioperatively (thoracotomy, CPB implementa-tion) but also postoperatively, with a 72 h follow-up after reinfusion.

Regarding the blood processing, the bleeding conditions in both models were adequate to collect enough blood to perform two cycle treatments by the device (first cycle volume of 700 mL and second of 500 mL). Both blood treatment and the subsequent reinfusion were per-formed in a very short time, compatible with the surgical time.

Animal clinical follow-up was limited to 72 hours +/- 4 hours postoperatively. This time covers the 24 hour period during which the acute complications related to transfusion are observed in the clinical practice [55]. A 72 h postoperative period was suitable to visualize the potential presence of large thrombi or organ infarction. These results are supported by the regression of the post-procedure inflammatory reaction seen through the decay of the neutro-philic leukocytosis after the first 24 hours.

The evaluated device demonstrated valuable performance related to its filtration technol-ogy. Heparin concentration in the treated blood was found below the regulatory threshold in both models (< 0.5 IU/mL) and heparin clearance was superior to 90% (97.4 ± 2.2%). Regard-ing the free hemoglobin removal, the AABB defined the threshold at 90% regardless of the free hemoglobin concentration [39]. In the present study, the removal of free hemoglobin was found to be superior to 90% (91.7 ± 3.5%), while the hemolysis rate in the treated blood was around 1.45%. Those data suggest that free hemoglobin was generated during the blood treat-ment but over 90% could be cleared. Hemolysis may also explain the lower removal perfor-mance of hemoglobin compared to the heparin washout (91.7% vs 97.4% for the free hemoglobin removal and the heparin washout, respectively). These results can be explained by the high hemolysis of porcine RBCs due to their fragility [56], while in human surgeries, hemolysis rate is about 1.3% [57]. The RBCs sensitivity to hemolysis has two consequences when compared to human blood treatment: hemolysis in the treated blood is higher and RBCs yield is lower. Indeed, in a preliminary *in vitro* study testing the system with human blood, RBC yield was over 88.1% and the final hemolysis was 0.12% [30]. A comparative study in swine with a system already available would allow confirmation of the fragility of porcine RBCs under these conditions and would rule out a specific impact of the i-SEP device. Hemo-lysis in the porcine model can be therefore considered as a worst-case scenario compared to

human blood. The higher hemolysis rate in the visceral model compared to the cardiac model is explained by the blood shedding, the coagulation activation during abdominal bleeding and the suction conditions (2.07 [0.55–2.42] % for the visceral model and 0.15 [0.1–0.3] % for the cardiac model). In the cardiac model, the blood is collected directly from the extra-corporeal circulation and does not come into contact with the extravascular medium. Despite the hemolysis rate higher than the 0.8% threshold (as defined by the European guidelines [58]) and the RBCs yield below 80%, the treated blood reached the 45% to 65% hematocrit reference range and its reinfusion was well tolerated by the animals. No clinical sign which might have been related to the reinfusion of free hemoglobin was observed. However, the small number of animals included in the study does not allow conclusions to be drawn about the safety related to the reinfusion of free hemoglobin.

The i-SEP device is the first autotransfusion system which allows in the same time, automatically and in a time compatible with the surgery, to concentrate not only red blood cells but also platelets. Indeed, the obtained treatment rate was higher than 80 mL/min, which is faster than the rate encountered by machines treating blood per centrifugation, even when using the emergency mode [59], and allowed the treatment of 500 mL to 700 mL diluted shed blood in less than 6 minutes. Moreover, platelet yield was always greater than 40% and higher than that obtained in the preliminary *in vitro* study of i-SEP using human blood [30]. Platelet concentration yields with the i-SEP device is higher than the ones that are usually observed with devices using centrifugation as a concentration method in *in vitro* and *in vivo* studies [59]. Reinfused blood included 83.3 ± 44.9% and 56.9 ± 5.6% of platelets collected in the blood collection reservoir in the visceral and cardiac model respectively. As expected, the platelet yield obtained after the blood treatment in the cardiac model is lower than that observed in the visceral model. That is attributable to the known deleterious effect of cardiopulmonary bypass on platelets [60–63] which therefore reduces the number of platelets to concentrate. In the visceral model, the presence of significant hemolysis in the collected blood produced red cells ghosts that may overestimate the platelet count. Some of the platelet yields were over 100% in the visceral model.

The reinfusion of platelets is considered as a potential prothrombotic risk. However, it was shown that the i-SEP autotransfusion system does not activate platelets during the treatment, and that platelets keep their ability to be activated after the treatment [30]. However, platelet activation was not evaluated in this animal study. Moreover, if platelets may be activated by the cardiopulmonary bypass in the cardiac model, the reinfusion of platelets during this study did not lead to any complication during the 72 h follow-up. No complication has been highlighted during the follow-up that could show a thrombogenic risk, in either model. After euthanasia, macroscopic examination of the different organs of interest and histological analysis did not reveal any evidence of thrombus formation. Moreover, even though the concentrated blood contains red blood cells and platelets, the coagulation power is annihilated because of the washing of coagulation factors in particular as described in a previous study [30] and as shown by the EXTEM and HEPTEM tests that demonstrated the inability of the treated blood to coagulate. However, the thrombotic risk was only evaluated during a postoperative period of 72 hours and therefore only the acute risks were assessed. Long-term follow-up on a larger number of individuals would allow conclusions to be drawn about safety, also regarding long-term risks.

The main limit of the study design was the descriptive nature of the results due to the limited number of animals per group that precluded any comparative statistical analysis. Moreover, the swine blood characteristics resulted in difficulties interpreting the hemolysis rate during blood treatment and the swine state of hypercoagulability required a partial anticoagulation with heparin in the visceral model that did not exactly replicate the clinical situation. As

mentioned above, the evaluation of the thrombotic risk would be improved with a longer follow-up, a larger number of individuals in each group and a more complete hemostasis and inflammation laboratory testing (platelet function, inflammatory proteins, white blood cell activity). It was chosen to use a controlled hemorrhagic model with moderate blood loss (less than 30% of the total blood volume) that allowed blood reinfusion without a need of complementary plasma transfusion. Therefore, this study did not replicate a massive hemorrhagic situation.

## Conclusions

This first *in vivo* study confirmed that this new autotransfusion system with an innovative filtration technology allowed to concentrate not only RBCs but also platelets, within a limited blood processing time. The concentrated blood did not have any pro-coagulant effect nor did cause any thrombotic effect after reinfusion to minipigs after cardiac or visceral bleeding. This autotransfusion device produced a concentrated blood ready for reinfusion in less than 6 minutes, with a high concentration of RBCs and platelets while removing efficiently heparin and free hemoglobin. Other studies should be implemented to compare the i-SEP device with other cell saver systems currently in use and evaluate clinical benefits of the transfusion of functional platelets during surgery.

## Supporting information

**S1 File.**
(DOCX)

## Author Contributions

**Conceptualization:** Benoit Decouture, Stéphane Chollet, Catherine Wittmann, Francis Gadrat, Alexandre Mansour, Patricia Forest-Villegas, Olivier Gauthier, Gwenola Touzot-Jourde.

**Data curation:** Kévin Schreiber, Benoit Decouture, Audrey Lafragette, Stéphane Chollet, Marine Bruneau, Maxence Nicollet, Olivier Gauthier, Gwenola Touzot-Jourde.

**Formal analysis:** Kévin Schreiber, Benoit Decouture, Audrey Lafragette, Stéphane Chollet, Marine Bruneau, Maxence Nicollet, Francis Gadrat, Gwenola Touzot-Jourde.

**Investigation:** Kévin Schreiber, Benoit Decouture, Audrey Lafragette, Stéphane Chollet, Marine Bruneau, Maxence Nicollet, Olivier Gauthier, Gwenola Touzot-Jourde.

**Methodology:** Benoit Decouture, Stéphane Chollet, Catherine Wittmann, Francis Gadrat, Alexandre Mansour, Patricia Forest-Villegas, Olivier Gauthier, Gwenola Touzot-Jourde.

**Project administration:** Catherine Wittmann, Patricia Forest-Villegas, Olivier Gauthier.

**Resources:** Alexandre Mansour.

**Supervision:** Francis Gadrat, Patricia Forest-Villegas, Olivier Gauthier, Gwenola Touzot-Jourde.

**Writing – original draft:** Kévin Schreiber.

**Writing – review & editing:** Benoit Decouture, Audrey Lafragette, Stéphane Chollet, Marine Bruneau, Maxence Nicollet, Catherine Wittmann, Francis Gadrat, Alexandre Mansour, Patricia Forest-Villegas, Olivier Gauthier, Gwenola Touzot-Jourde.

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
