## [Decision Letter · Decision Letter 0]

24 Aug 2021

PONE-D-21-23852

A novel autotransfusion device saving erythrocytes and platelets by filtration used in a 72h survival swine model of controlled blood loss: perioperative hematologic and coagulation assessments, salvaged blood characteristics and system performance

PLOS ONE

Dear Dr. Touzot-Jourde,

Thank you for submitting your manuscript to PLOS ONE. After careful consideration, we feel that it has merit but does not fully meet PLOS ONE’s publication criteria as it currently stands. Therefore, we invite you to submit a revised version of the manuscript that addresses the points raised during the review process.

We look forward to receiving your revised manuscript.

Kind regards,

Ahmet Emre Eşkazan

Academic Editor

PLOS ONE

Journal Requirements:

"Dr. Benoit Decouture (BD) is currently employed as project manager by i-SEP (Nantes, France).

Stéphane Chollet (SC) is currently employed as technical director by i-SEP (Nantes, France).

Marine Bruneau (MB) was employed as research and development engineer by i-SEP (Nantes, France).

Maxence Nicollet (MN) was employed as a technician by i-SEP (Nantes, France).

Dr. Catherine Wittman (CW) was employed as a compliance and clinical affairs deputy head by i-SEP (Nantes, France).

Dr. Francis Gadrat (FG) is currently employed as medical director by i-SEP (Nantes, France).

Dr. Patricia Forest-Villegas (PVF) is currently employed as scientific director by i-SEP (Nantes, France)."

Reviewers' comments:

Reviewer's Responses to Questions

**Comments to the Author**

1. Is the manuscript technically sound, and do the data support the conclusions?

Reviewer #1: Yes

Reviewer #2: Yes

Reviewer #3: Yes

2. Has the statistical analysis been performed appropriately and rigorously? 

Reviewer #1: Yes

Reviewer #2: Yes

Reviewer #3: Yes

3. Have the authors made all data underlying the findings in their manuscript fully available?

Reviewer #1: Yes

Reviewer #2: Yes

Reviewer #3: Yes

4. Is the manuscript presented in an intelligible fashion and written in standard English?

Reviewer #1: Yes

Reviewer #2: Yes

Reviewer #3: Yes

5. Review Comments to the Author

Reviewer #1: In the manuscript, the authors evaluated the performances of a new autotransfusion system which can collect bloods and platelets simultaneously using a swine model. The topic is both interesting and important. There are several minor problems need to be improved:

1. It is better if the autotransfusion system and the study design can be depict using a schematic diagram.

2. Obviously the greatest advantage of this device is it can collect bloods and platelets simultaneously, can you simply explain how it works?

Reviewer #2: This study described a new swine model of surgical blood loss that could be used to evaluate a cell saver or autotransfusion system. The study also described a new autotransfusion system. In some ways, this reads like two separate manuscripts. The manuscript is very lengthy and contains a lot of data and text. It also reads like two manuscripts have been combined.

Concerning the part on the new autotransfusion system, the tables and figures contain all of the important data, but the text of the results section does not mention some important information such as the recovery of platelets. This information is, however, mentioned in the discussion section.

Reviewer #3: The authors investigated on the performance of a new intra-operative autotransfusion system, i-Sep device, using two different models of controlled bleeding. It is a new system for the intraoperative blood salvage, which recovers not only red blood cells, but also platelets and leukocytes. The main point to consider is if transfusion of recovered platelets is effective in terms of hemostasis and if it can reduce transfusion of allogeneic platelets, and if they do not cause harm to the patient. Also, the consequences of recovering and transfusing leukocytes needs consideration.

Autologous blood transfusion is an important alternative in terms of patient blood management, and the intraoperative cell salvage is advantageous since it does not cause burden to the patients in collecting autologous blood pre-operatively.

The paper is quite confusing because there are two objectives in the study, first to develop pig models of controlled cardiac and visceral bleeding, and second to test the effectiveness of i-Sep as an autotransfusion system. It is important that both objectives are clearly stated, and conclusions are drawn for both the objectives.

It would be interesting to see the i-Sep results in comparison with an existing system, such as the Auto Saver, using established animal models.

There are many points that must be addressed before the paper can be considered for publication.

Major points:

1. What are the advantages of the established animal models compared to the existing ones, and how can they be applied in the new future?

2. In the cardiac model, what was the priming solution used for the cardio-thoracic bypass? A priming volume of 1250mL was used, which may affect the general results.

3. I understand on the importance of transfusing platelets together with RBC, especially in patients with massive bleeding, but I am not sure if platelets recovered from the cardio-thoracic bypass are effective enough to be transfused. It is very well-known that the cardio-thoracic bypass activates platelets, resulting in disturbed platelet function, as well as low platelet counts, which is compatible with their results of lower platelet yield in the cardiac model. There is no data, however, showing the number of platelets recovered from the cardio-thoracic bypass, nor on their in vitro activation or functional status. These data should be shown, at least for the cardiac model.

4. What is the advantage of recovering and transfusing leukocytes together with RBC and platelets? The universal leukoreduction has been implemented in various countries as a strategy to prevent alloimmunization and prevent non-hemolytic transfusion reactions. Since the transfused leukocytes are autologous, there is no risk of alloimmunization, and since autologous blood is not stored, I believe the risk of cytokine accumulation that occurs during storage would not be a problem, but it is necessary to compare leukoreduced and non-leukoreduced products, or at least discuss on the advantages/disadvantages of transfusing autologous leukocytes.

5. In both models, the authors performed blood sampling before surgery (T0), then at the end of transfusion (T1) and post-operatively. Did the authors perform blood sampling after the bleeding, that is between T0 and T1? What was the hemoglobin trigger to start the transfusion? How much did the hemoglobin levels drop after bleeding of more than 20% of the circulating blood volume? What were the platelet values before and after the bleeding, and how did they recover after the transfusion? This information is essential to conclude if their models are useful for the evaluation of transfusion practices in an animal model. Especially in the cardiac model, how much platelets were lost during the cardio-thoracic bypass, how was the function and activation status of the recovered platelets? Was the effect of blood transfusion poorer compared to the visceral model?

6. The massive transfusion protocol recommends transfusion of RBC: PLT: plasma at a 1:1:1 ratio, and recently, the pre-hospital transfusion of whole blood is being performed especially in trauma centers in the US, with very promising results. In the present study, RBC and platelets are transfused without plasma, that means without coagulation factors. What is the consequence of this?

7. Describe the limitations of the study at the end of Discussion.

Minor points:

1. Figure legends should be removed from the text and described in a separate section.

2. In line 48, 248, 271-2, 276, 364, 555-9, 563 (Table 4) and 631: There are descriptions of “UI” and “IU”. It should be expressed as “IU”.

3. In lines 133 and 681: The equipment is described as “I-Sep” and “i-Sep”. Which one is correct?

4. In line 173: “done just prior morning feeding” should be “done just prior to morning feeding”

5. In line 254: “insure” should be “ensure”

6. In line 322: “Iso” should be “ISO”.

7. In lines 322-324: There is need to close the quotation mark (“)

8. In lines 366, 441, 454, 563 (Table 4),Fig 1(B) and Fig 2(B): The units of hemoglobin are differently described, such as “(500-1000 mg/dL)” “Hemoglobin in mg/L” “Total hemoglobin concentration (g/dL)” “Hemoglobin concentration (g/L)” . Please check.

9. In line 360: “autotransfusion system washing” should be “autotransfusion system in washing”

10. In line 382: “Hematocrite” should be “Hematocrit”

11. In lines 440-442: There is need to close the quotation mark (“)

12. In lines 480-483: The descriptive statistics should be given for the platelet counts (median [min-max])

13. In lines 485-489 and Fig.4 (A): The data of 4 animals are shown and described in the visceral model. I believe there are 5 animals in this model. Please check and correct.

14. In line 503: “compared before” should be “compared to before”

15. In lines 506-507: In Table 1, some values are expressed in non-bold characters.

16. In line 530: In Table 3, should “Amplitude 6 minutes” be expressed in bold characters?

17. In lines 556-557and 563-564: The descriptions of heparin concentrations of “0.14 [0.00-0.36] UI/mL” from visceral model and “0.46 [0.44-0.60] UI/mL” from the cardiac model do not match the values described in Table 3 “0.16 ± 0.19” in visceral model and “0.50 ± 0.08” in cardiac model. Similarly, the hemolysis rate described in lines 652-653 does not match that described in Table 3.

18. In line 561: From Table 4, the criterium of RBC yield was not achieved, and there was high hemolysis, so it would be better to describe it correctly as follows: “performance parameters (heparin clearance, free hemoglobin washout, hemolysis rate, RBC yield, hematocrit and hemoglobin concentration) exceeded set objectives, except for RBC yield and hemolysis rate (Table 4).”

19. In line 564: I suppose the data in Table 4 is shown as mean ± SD, and not as “median” as described. Please check.

20. In line 569: “for both model” should be “for both models”

21. In Figure 4: I suppose there were 5 animals in the visceral model, but the figure shows data from 4 animals. Please check.

22. In line 600: “they concluded was that” should be “they concluded that”

23. In line 604: “The porcine model appears as” would be better “The porcine model seems to be”

24. In line 611: “as close as possible clinical indications” would be better “as close as possible to the clinical settings”

25. In line 613: “thoracotomy and blood treatment of the remaining blood in the CPB” would be better “thoracotomy and treatment of the remaining blood in the CPB”

26. In line 613-614: “(ii) a abdominal visceral model” should be “(ii) an abdominal visceral model”

27. In line 619: “blood treatment processing” should be “blood processing”

28. In line 620: “two cycle treatment” should be “two cycle treatments”

29. In line 622: “compatible with the surgery” would be better “compatible with the surgical time”

30. In line 625: “in human medicine” would be better “in the clinical practice”

31. In lines 636-637: “but was cleared over 90%” would be better “but over 90% could be cleared”

32. In lines 637-638: “This hemolysis phenomenon also explains the lesser performance of free hemoglobin removal compared to the heparin washout” would be better “Hemolysis may also explain the lower removal performance of hemoglobin compared to the heparin washout”

33. In lines 640-643: the sentence “Preliminary results with the present system…” is difficult to understand. Elaborate better

34. In line 644: “hemolysis in the treated blood was higher” should be “hemolysis in the treated blood is higher”

35. In line 645: “in vitro study on human blood testing the system” would be better “in vitro study testing the system with human blood”

36. In line 650: “compared to human blood treatment” should be “compared to human blood”

37. In line 651-652: “explained by blood shedding, coagulation activation during abdominal bleeding and suction conditions” should be “explained by the blood shedding, the coagulation activation during abdominal bleeding and the suction conditions”

38. In line 655-656: “Despite the hemolysis superior to the 0.8% threshold and RBCs yield below 80%” should be “Despite the hemolysis rate higher than the 0.8% threshold and the RBCs yield below 80%”

39. In lines 658-659: “free hemolysis” should be “free hemoglobin”

40. In line 663: “the surgery the concentration of red blood cells” should be “the surgery, to concentrate not only red blood cells”

41. In lines 667-670: “platelet yield always greater than 40% and greater than the one obtained in the preliminary i-Sep study on in vitro human blood” would be better “platelet yield was always greater than 40% and higher than that obtained in the preliminary in vitro study of i-Sep using human blood”

42. In line 678: “Some of the platelet yield over 100%” should be “Some of the platelet yields were over 100%”

43. In line 686: “whichever the animal model” would be better “in either model”

44. In lines 687-688: “any thrombus in any of them” would be better “any evidence of thrombus formation”

45. In line 691: “the incapacity of the treated blood to coagulate” would be better “the inability of the treated blood to coagulate”

46. In lines 699-700: “allowed not only RBCs concentration but also platelet one” would be better “allowed to concentrate not only RBCs but also platelets”

47. In line 701: “pro-coagulant effect nor cause” should be “pro-coagulant effect nor did cause”

48. In line 702: “cardiac or visceral” should be “cardiac or visceral bleeding”

6. PLOS authors have the option to publish the peer review history of their article (what does this mean?). If published, this will include your full peer review and any attached files.

Reviewer #1: No

Reviewer #2: No

Reviewer #3: No

---

## [Author Response · Author response to Decision Letter 0]

2 Oct 2021

Dear Editor and Reviewers,

Please find below our responses to the reviewer comments and suggestion for the manuscript ONE-D-21-23852 entitled “A novel autotransfusion device saving erythrocytes and platelets by filtration used in a 72 h survival swine model of controlled blood loss: perioperative hematologic and coagulation assessments, salvaged blood characteristics and system performance” 

To ease reading our answers, the text is displayed in blue right below the point raised or question asked buy the Reviewers

Journal Requirements:

We have used the PLOS ONE template to submit our manuscript and checked its conformity to PLOS ONE style requirements.

"Dr. Benoit Decouture (BD) is currently employed as project manager by i-SEP (Nantes, France).

Stéphane Chollet (SC) is currently employed as technical director by i-SEP (Nantes, France).

Marine Bruneau (MB) was employed as research and development engineer by i-SEP (Nantes, France).

Maxence Nicollet (MN) was employed as a technician by i-SEP (Nantes, France).

Dr. Catherine Wittman (CW) was employed as a compliance and clinical affairs deputy head by i-SEP (Nantes, France).

Dr. Francis Gadrat (FG) is currently employed as medical director by i-SEP (Nantes, France).

Dr. Patricia Forest-Villegas (PVF) is currently employed as scientific director by i-SEP (Nantes, France)."

We have implemented the changes and confirmed in the cover letter that the identified competing interests have not alter our ability to share data and materials from the study.

Reviewers' comments:

Reviewer's Responses to Questions

Comments to the Author

1. Is the manuscript technically sound, and do the data support the conclusions?

Reviewer #1: Yes

Reviewer #2: Yes

Reviewer #3: Yes

2. Has the statistical analysis been performed appropriately and rigorously? 

Reviewer #1: Yes

Reviewer #2: Yes

Reviewer #3: Yes

3. Have the authors made all data underlying the findings in their manuscript fully available?

Reviewer #1: Yes

Reviewer #2: Yes

Reviewer #3: Yes

4. Is the manuscript presented in an intelligible fashion and written in standard English?

Reviewer #1: Yes

Reviewer #2: Yes

Reviewer #3: Yes

5. Review Comments to the Author

Reviewer #1: In the manuscript, the authors evaluated the performances of a new autotransfusion system which can collect bloods and platelets simultaneously using a swine model. The topic is both interesting and important. There are several minor problems need to be improved:

1. It is better if the autotransfusion system and the study design can be depict using a schematic diagram.

A schematic diagram has been added to help visualizing the experimental steps (hemorrhage, transfusion, post-operative care) and times of blood samplings.

2. Obviously the greatest advantage of this device is it can collect bloods and platelets simultaneously, can you simply explain how it works?

The SAME device (Smart Autotransfusion for Me; i-SEP, France) was designed as an innovative filtration-based autotransfusion device able to salvage and wash both red blood cells and platelets. This new autotransfusion system integrates a hollow fiber filtration technology, comparable to the filters used for plasmapheresis or for ultrafiltration during cardiopulmonary bypass. Using a combination of washing and filtration of salvage blood, the device allows the concentration of red blood cells and platelets within the concentrated blood product, as well as the removal of heparin, free hemoglobin, coagulation factors, and inflammatory mediators such as complement proteins (already published in Mansour et al., Anesthesiology, 2021).

The SAME device is a medical device consisting of reusable equipment and disposable consumables (See digital content 1, http://links.lww.com/ALN/C621). The device innovative technology and process are described in the following patents: PCT/FR2018/053500 published as WO 2019/129973 on July 4, 2019 (corresponding to U.S. application no. 16/958,473); PCT/FR2018/053501 published as WO 2019/129974 on July 4, 2019 (corresponding to U.S. application no. 16/958,458); and PCT/ FR2020/051115 published as WO 2020/260836 on December 30, 2020. Consumables include a dual-lumen suction line (allowing both collection and anticoagulation of shed blood), a blood collection reservoir (including a 40-μm filter), and a treatment set. The treatment set includes tubing, a polyethersulfone hollow fiber cartridge that separates the blood cells from the plasma, a compliant blood treatment bag that ensures the blood washing, a waste bag that receives the plasma and contaminants, and a reinfusion bag that stores the filtered, washed, and concentrated blood cells (See digital content 2, http:// links.lww.com/ALN/C622). The reusable equipment is an electromedical medical device composed by several systems required for blood circulation and continuous measurements, including a continuous in-line hematocrit monitor. The i-SEP device is associated with a specific software to drive the different steps of the device installation and blood treatment.

During clinical use, the first stage of cell salvage with the i-SEP device is the collection of shed blood from the surgical field by the dual-lumen suction line, allowing the anticoagulation of shed blood by a heparinized saline drip. The shed blood is collected in the collection reservoir in which it undergoes a first filtration by the included 40-μm filter, allowing the removal of bone debris and microaggregates before blood treatment by the device. 

Then the treatment set is filled with anticoagulated salvaged blood transferred from the blood collection reservoir when a sufficient volume is collected. During the treatment phase, the blood is processed by the i-SEP device, with simultaneous filtration and washing (Supplemental Digital Content 2, http://links.lww.com/ALN/C622). The volume of the treatment set (300 to 1,000 ml thanks to the compliant treatment bag) limits the amount of collected blood that can be processed in one time, hence defining a treatment cycle. Several simultaneous steps then constitute the innovative i-SEP process: wash solution (normal saline) is pumped into the treatment set; diluted salvaged blood circulates within the treatment set between the treatment bag and the polyethersulfone hollow fiber to allow microfiltration to occur; and fluid is continuously discarded from the treatment circuit into the waste bag through the effluent line. Once the continuously monitored hematocrit reaches the prespecified target, the device automatically transfers the processed blood from the treatment set into the reinfusion bag.

The reference of the device description (Mansour et al., Anesthesiology, 2021) has been added in the Materials and Methods.

Reviewer #2: This study described a new swine model of surgical blood loss that could be used to evaluate a cell saver or autotransfusion system. The study also described a new autotransfusion system. In some ways, this reads like two separate manuscripts. The manuscript is very lengthy and contains a lot of data and text. It also reads like two manuscripts have been combined.

The objectives of the study have been reformulated at the end of the introduction to better explain the rationale of the study. Indeed, the aim of this study was twofold and we specified it in the Introduction of the manuscript. The dual objective of this study was first to develop a swine surgical model of controlled blood loss with a 72 h post-operative follow-up that would reproduce the best possible clinically relevant surgical events, and then to evaluate the use of the autotransfusion device in this swine model. The afterward aim is to use this controlled blood loss surgical model to pursue the device development and its preclinical evaluation in a comparative study with a cell saver already on the market.

Concerning the part on the new autotransfusion system, the tables and figures contain all of the important data, but the text of the results section does not mention some important information such as the recovery of platelets. This information is, however, mentioned in the discussion section.

The result section corresponding to the tables and figures on the autotransfusion performance has been rewritten.

Reviewer #3: The authors investigated on the performance of a new intra-operative autotransfusion system, i-Sep device, using two different models of controlled bleeding. It is a new system for the intraoperative blood salvage, which recovers not only red blood cells, but also platelets and leukocytes. The main point to consider is if transfusion of recovered platelets is effective in terms of hemostasis and if it can reduce transfusion of allogeneic platelets, and if they do not cause harm to the patient. Also, the consequences of recovering and transfusing leukocytes need consideration.

Transfusion of recovered platelets

Reinfusion of platelets is indeed a crucial question as it could induce prothrombotic events in the patient blood circulation. 

Thus, in our previous study on human blood (Mansour et al., Anesthesiology, 2021), we evaluated the platelet activation after blood treatment and showed that platelets had a low P-selectin expression after treatment (10.8%) and kept a strong response to thrombin-activating peptide. These results demonstrate that platelets were not activated during the treatment, even partially or reversibly, as they are not refractory to activation.

Moreover, in the HAS (Haute Autorité de Santé) and the ANSM (Agence Nationale de Sécurité du Médicament et des produits de santé) document about the good practice recommendation of platelet transfusion (“Transfusion de plaquettes: produits, indications” HAS and ANSM, argumentaire scientifique, October 2015), moderated disturbances of the platelet activity are detected in concentrated platelets from apheresis. Notably, the study of Metcalfe et al. shows between 25% and 30% of p-selectin positive platelets in apheresis platelets (Metcalfe et al., Br J Haematol, 1997). Recent studies show similar results (Bontekoe et al, Vox Sanguinis, 2018; Sperling et al, Hematology, 2018). Percentage of p-selectin positive platelets was between 15% and 25% after one day-storage, 39% after two days-storage and above 60% after five days-storage. In the concentrated blood obtained with the i-Sep Autotransfusion System, the percentage of p-selectin positive platelets (i.e. activated platelets by the treatment, 10.8%) was lower than those in the study of Metcalfe et al. Then, the results show that the platelets obtained with the i-Sep Autotransfusion System are compatible with transfusion.

Transfusion of recovered leukocytes

Leukocyte transfusion is also a crucial question as activated leukocytes could induce strong pro-inflammatory events which should be avoided in a recovering patient. 

However, although the other autotransfusion devices claim that leukocytes are eliminated during blood treatment, literature showed high white blood cell recovery: the Sorin Xtra device white blood cell recovery is between 55% and 70% (Overdevest et al., Perfusion, 2012), the Fresenius Kabi devices white blood cell recovery are 65.6% for the CATSmart and 81.1% for the C.A.T.Splus (Alberts et al., J Extra Corpor Technol, 2017).

Regarding the i-SEP device, the previous study showed a 93.0% white blood cell recovery (Mansour et al., Anesthesiology, 2021). As the recovery was very high, white blood cell death or activation was evaluated. Leukocyte viability in pretreated blood (collection reservoir and treatment bag) was 97.6% for the treatment first cycle and 97.7% for the second cycle. After treatment, the leukocyte viability was changed by less than 1%. Regarding basal activation of leukocytes in the pretreated blood, respectively, for the first and second cycles: the percentages of HLA-DR positive/CD4- positive cells were 4.4% and 4.4%; the percentages of HLA-DR positive/CD8-positive cells were 13.2% and 12.7%; CD64 surface expression levels on neutrophils were 1,420 and 1,463; and CD64 surface expression levels on monocytes were 11,122 and 12,219. Cell recovery was not associated with significant leukocyte activation, either regarding CD4-positive cells (−0.1% for the first cycle; 0.0% for the second cycle) or CD8-positive cells (−3.2% for the first cycle; −2.0% for the second cycle). Last, blood treatment did not induce any significant increase in CD64 surface expression in post-treated compared to pretreated blood for neutrophils (8, for the first cycle; −30 for the second cycle) or monocytes (1,905 for the first cycle; 794 for the second cycle).

In conclusion, obtained results in the previous study on human blood showed that the leukocyte reinfusion from the i-SEP device comparable to the other autotransfusion devices already in use. 

Autologous blood transfusion is an important alternative in terms of patient blood management, and the intraoperative cell salvage is advantageous since it does not cause burden to the patients in collecting autologous blood pre-operatively.

The paper is quite confusing because there are two objectives in the study, first to develop pig models of controlled cardiac and visceral bleeding, and second to test the effectiveness of i-Sep as an autotransfusion system. It is important that both objectives are clearly stated, and conclusions are drawn for both the objectives.

The objectives of the study have been reformulated at the end of the introduction to better explain the rationale of the study. Indeed, the aim of this study was twofold and we specified it in the Introduction of the manuscript. The dual objective of this study was first to develop a swine surgical model of controlled blood loss with a 72 h post-operative follow-up that would reproduce the best possible clinically relevant surgical events,, and then to evaluate the use of the autotransfusion device in this swine model. 

It would be interesting to see the i-Sep results in comparison with an existing system, such as the Auto Saver, using established animal models.

This suggestion is indeed very relevant. A comparative study with a cell saver already on the market is indeed the next step. 

There are many points that must be addressed before the paper can be considered for publication.

Major points:

1. What are the advantages of the established animal models compared to the existing ones, and how can they be applied in the new future?

The main indication of using an autotransfusion device in humans are surgical procedures with a potential for massive bleeding and cardiac surgery using extracorporeal circulation. The experiments intended to create with swine models a clinical situation as close as possible to the surgical situations in human operating room. 

Cardiac model: Collecting diluted blood from the bypass circuit after patient blood volume adjustment at the end of the extracorporeal circulation followed retransfusion to the animal was therefore chosen to mimic the clinical situation.

Visceral model : Concerning the splenic model, the main objectives of the model were:

- to develop a model of visceral bleeding in which shed blood would be in contact with the visceral serosa, in order to reproduce a clinical situation to reproduce tissue bleeding in which blood contacts surrounding extravascular tissue;

- to create a visceral injury that allow massive bleeding while still being controllable by the bleeding organ resection;

- to have a survival model allowing to document hematological data during the postoperative period.

Therefore, classical controlled hemorrhagic models using venous or arterial blood aspiration even with an organ injury (liver lobe resection for example) were not deemed appropriate. Compared to liver trauma that are used in models of uncontrolled hemorrhage, the splenic hemorrhage was found to occur sufficiently rapidly and profusely to be equivalent to a liver trauma model while surgical hemostasis of the organ was found easy and predictable, which is not the case for the liver. The aim was also to be able to assess survival from the hemorrhage and retransfusion without having the confounding factors of a major traumatic vital organ resection.

2. In the cardiac model, what was the priming solution used for the cardio-thoracic bypass? A priming volume of 1250mL was used, which may affect the general results.

The priming volume chosen was a Ringer Lactate solution. Blood gas analysis during the on bypass pump time showed metabolic stability (normal pH, no to slight elevation of lactate but less than 4 mmol/l, stable electrolytes). It was chosen to minimally interfere (apart from the dilutional effect) with coagulation compared to a mixed priming solution that would contain colloids. The dilutional effect was more visible on the plasma protein concentration (by a decrease of about 20%) than on the hematocrit that inconstantly showed no to minor decrease (probable splenic contraction releasing RBC). 

The information was added into the appropriate section.

3. I understand on the importance of transfusing platelets together with RBC, especially in patients with massive bleeding, but I am not sure if platelets recovered from the cardio-thoracic bypass are effective enough to be transfused. It is very well-known that the cardio-thoracic bypass activates platelets, resulting in disturbed platelet function, as well as low platelet counts, which is compatible with their results of lower platelet yield in the cardiac model. There is no data, however, showing the number of platelets recovered from the cardio-thoracic bypass, nor on their in vitro activation or functional status. These data should be shown, at least for the cardiac model.

It has been well described in the literature that the CBP is associated with platelet dysfunction, platelet activation and platelet loss (Muriithi EW et al., The effects of heparin and extracorporeal circulation on platelet counts and platelet microaggregation during cardiopulmonary bypass. J Thorac Cardiovasc Surg. 2000 Sep;120(3):538-43; Tamari Y et al., Functional changes in platelets during extracorporeal circulation. Ann Thorac Surg. 1975 Jun;19(6):639-47.). However, it has been shown that only a proportion of platelets are activated and lost, and that around 50% of the platelets remain in the circulating blood (Griffin BR, Bronsert M, Reece TB, et al. Thrombocytopenia After Cardiopulmonary Bypass Is Associated With Increased Morbidity and Mortality. Ann Thorac Surg. 2020). Platelet concentrations in the blood collection reservoir (TR) were 54 (16-83) 103/µL and 157 (115-192) 103/µL for the visceral and the cardiac model respectively. These data have been added in the text and in the Table 4.

Platelet function can also be affected by CPB and be reduced by up to 50% (Agarwal S et al., Pre- and Post-Bypass Platelet Function Testing With Multiple Electrode Aggregometry and TEG Platelet Mapping in Cardiac Surgery. J Cardiothorac Vasc Anesth. 2015 Oct;29(5):1272-6). Regarding platelets activity in the blood recovered from the extracorporeal circulation, the blood dilution in the collection reservoir did not allow to measure platelet (with platelet ROTEM) nor coagulation activity (ROTEM coagulation tests). However, after blood transfusion and all along the post-operative follow-up, platelet activity and coagulation tests on the animal blood samples were normal. 

4. What is the advantage of recovering and transfusing leukocytes together with RBC and platelets? The universal leukoreduction has been implemented in various countries as a strategy to prevent alloimmunization and prevent non-hemolytic transfusion reactions. Since the transfused leukocytes are autologous, there is no risk of alloimmunization, and since autologous blood is not stored, I believe the risk of cytokine accumulation that occurs during storage would not be a problem, but it is necessary to compare leukoreduced and non-leukoreduced products, or at least discuss on the advantages/disadvantages of transfusing autologous leukocytes.

Leukocyte transfusion is also a crucial question as activated leukocytes could induce strong pro-inflammatory events which should be avoided in a recovering patient. 

However, although the other autotransfusion devices claim that leukocytes are eliminated during the blood treatment, literature showed high white blood cell recovery: the Sorin Xtra device white blood cell recovery is between 55% and 70% (Overdevest et al., Perfusion, 2012), the Fresenius Kabi devices white blood cell recovery are 65.6% for the CATSmart and 81.1% for the C.A.T.Splus (Alberts et al., J Extra Corpor Technol, 2017).

Regarding the i-SEP device, the previous study showed a 93.0% white blood cell recovery (Mansour et al., Anesthesiology, 2021). As the recovery was very high, white blood cell death or activation was evaluated. Leukocyte viability in pretreated blood was 97.6% for the first cycle and 97.7% for the second cycle. After the treatment, the leukocyte viability was changed by less than 1%. Regarding basal activation of leukocytes in the pretreated blood, respectively, for the first and second cycles: the percentages of HLA-DR positive/CD4- positive cells were 4.4% and 4.4%; the percentages of HLA-DR positive/CD8-positive cells were 13.2% and 12.7%; CD64 surface expression levels on neutrophils were 1,420 and 1,463; and CD64 surface expression levels on monocytes were 11,122 and 12,219. Cell recovery was not associated with significant leukocyte activation, either regarding CD4-positive cells (−0.1% for the first cycle; 0.0% for the second cycle) or CD8-positive cells (−3.2% for the first cycle; −2.0% for the second cycle). Last, blood treatment did not induce any significant increase in CD64 surface expression in post-treated compared to pretreated blood for neutrophils (8, for the first cycle; −30 for the second cycle) or monocytes (1,905 for the first cycle; 794 for the second cycle). Thus, the absence of leukocyte activation strongly annihilates the risk of cytokine secretion and accumulation.

In conclusion, obtained results in the previous study on human blood showed that the leukocyte reinfusion from the i-SEP device comparable to the other autotransfusion devices already in use. 

5. In both models, the authors performed blood sampling before surgery (T0), then at the end of transfusion (T1) and post-operatively. Did the authors perform blood sampling after the bleeding, that is between T0 and T1? What was the hemoglobin trigger to start the transfusion? How much did the hemoglobin levels drop after bleeding of more than 20% of the circulating blood volume?

Due to the repeated sampling times during the procedure and the 72 h follow-up for hematology and coagulation testing, we had to refine the numbers of samples to reduce the total volume of blood taken. As we had some prior data on hematocrit and plasma proteins while designing the visceral model, we felt that we didn’t need this extra information. It is however planned for the next study that will focus on the benefit of the transfusion compared to the other autotransfusion devices already on the market. 

In preliminary experiments and in the present study, we saw that the surgically triggered splenic injury induced a splenic contraction (visible during the surgery). We also used a minimal fluid resuscitation strategy during the blood loss (enough IV Ringer lactate solution to maintain mean arterial blood pressure around 60 mmHg) to avoid hemodilution. The animals lost between 600 and 900 mL of blood and they received between 1L and 1,5 L of Ringer lactate solution. The combination of both events resulted in stable hematocrit and hemoglobin during bleeding and retransfusion in preliminary experiments.

What were the platelet values before and after the bleeding, and how did they recover after the transfusion? This information is essential to conclude if their models are useful for the evaluation of transfusion practices in an animal model. Especially in the cardiac model, how much platelets were lost during the cardio-thoracic bypass, how was the function and activation status of the recovered platelets? Was the effect of blood transfusion poorer compared to the visceral model?

As mentioned for the previous question, due to the repeated sampling times during the procedure and the 72 h follow-up for hematology and coagulation testing, we had to refine the numbers of samples to reduce the total volume of blood taken. During this study, we did not focus on the direct effect of transfusion (just before and just after the transfusion). This evaluation is however planned for the next study that will focus on the benefit of the transfusion compared to the other autotransfusion devices already on the market. 

Data in the supplemental Table S1 allows to compare platelet count before and after surgery. Both models showed a very moderate decrease of the platelet count after the surgery (TP) compared to before (T0).

Regarding the platelet activity after the surgery, we also performed FIBTEM tests when using the ROTEM. FIBTEM is the same test as the EXTEM with a strong platelet inhibitor. Then, the difference between the EXTEM test and the FIBTEM test is the platelet contribution to the EXTEM clot formation. The Maximum Clot Firmness (MCF) is the evaluated parameter.

EXTEM MCF - FIBTEM MCF (mm) Model Parameter T0 TP T2-6 T6-12 T12-24 T24-36 T36-48 T48-60 T60-72

 Cardiac (n=4) Median 53 57 53,5 49,5 50 46 44,5 43,5 36,5

 Minimum 48 54 51 46 48 44 37 30 16

 Maximum 53 60 54 52 55 48 57 46 45

 Visceral (n=5) Median 49 53 51 50 50 46 45 47 43

 Minimum 40 49 38 40 38 35 42 37 34

 Maximum 55 54 57 53 51 48 51 52 51

Results show that platelets equally contributed to clot formation after the surgery (TP) when compared to before the surgery (T0). This way of evaluating platelet contribution to clot formation is not a classical way, thus these results were not added to the manuscript. However, platelet functionality will be the focus in the next studies. 

6. The massive transfusion protocol recommends transfusion of RBC: PLT: plasma at a 1:1:1 ratio, and recently, the pre-hospital transfusion of whole blood is being performed especially in trauma centers in the US, with very promising results. In the present study, RBC and platelets are transfused without plasma, that means without coagulation factors. What is the consequence of this?

This comment is very interesting as the blood dilution in the circulation without plasma and coagulation factor transfusion could lead to a prothrombotic state.

However, the protocol performed in this study does not reproduce a massive bleeding nor a massive transfusion case. The volume of the blood and anticoagulation solution mixture in the collection reservoir corresponded to a minimal of 20% and a maximum of 30% of the estimated animal blood volume. We anticipated that this volume would not impact the hemostasis of the animals. This expectation was confirmed by the ROTEM assays that showed unchanged clotting time (CT) after the surgery (TP) when compared to before the surgery (T0). 

We are currently planning a high bleeding volume model in which we plan to transfuse plasma volumes equivalent to washed plasma to avoid thrombotic events due to plasma dilution.

7. Describe the limitations of the study at the end of Discussion.

The limitations of the study have been rewritten and pooled at the end of the discussion

Minor points:

We have incorporated the appropriate changes into the manuscript following the minor comments that are listed below.

1. Figure legends should be removed from the text and described in a separate section.

2. In line 48, 248, 271-2, 276, 364, 555-9, 563 (Table 4) and 631: There are descriptions of “UI” and “IU”. It should be expressed as “IU”.

3. In lines 133 and 681: The equipment is described as “I-Sep” and “i-Sep”. Which one is correct?

4. In line 173: “done just prior morning feeding” should be “done just prior to morning feeding”

5. In line 254: “insure” should be “ensure”

6. In line 322: “Iso” should be “ISO”.

7. In lines 322-324: There is need to close the quotation mark (“)

8. In lines 366, 441, 454, 563 (Table 4),Fig 1(B) and Fig 2(B): The units of hemoglobin are differently described, such as “(500-1000 mg/dL)” “Hemoglobin in mg/L” “Total hemoglobin concentration (g/dL)” “Hemoglobin concentration (g/L)” . Please check.

9. In line 360: “autotransfusion system washing” should be “autotransfusion system in washing”

10. In line 382: “Hematocrite” should be “Hematocrit”

11. In lines 440-442: There is need to close the quotation mark (“)

12. In lines 480-483: The descriptive statistics should be given for the platelet counts (median [min-max]) � Added in the supplemental table 4.

13. In lines 485-489 and Fig.4 (A): The data of 4 animals are shown and described in the visceral model. I believe there are 5 animals in this model. Please check and correct.

14. In line 503: “compared before” should be “compared to before”

15. In lines 506-507: In Table 1, some values are expressed in non-bold characters.

16. In line 530: In Table 3, should “Amplitude 6 minutes” be expressed in bold characters?

17. In lines 556-557and 563-564: The descriptions of heparin concentrations of “0.14 [0.00-0.36] UI/mL” from visceral model and “0.46 [0.44-0.60] UI/mL” from the cardiac model do not match the values described in Table 3 “0.16 ± 0.19” in visceral model and “0.50 ± 0.08” in cardiac model. Similarly, the hemolysis rate described in lines 652-653 does not match that described in Table 3.

18. In line 561: From Table 4, the criterium of RBC yield was not achieved, and there was high hemolysis, so it would be better to describe it correctly as follows: “performance parameters (heparin clearance, free hemoglobin washout, hemolysis rate, RBC yield, hematocrit and hemoglobin concentration) exceeded set objectives, except for RBC yield and hemolysis rate (Table 4).”

19. In line 564: I suppose the data in Table 4 is shown as mean ± SD, and not as “median” as described. Please check.

20. In line 569: “for both model” should be “for both models”

21. In Figure 4: I suppose there were 5 animals in the visceral model, but the figure shows data from 4 animals. Please check.

22. In line 600: “they concluded was that” should be “they concluded that”

23. In line 604: “The porcine model appears as” would be better “The porcine model seems to be”

24. In line 611: “as close as possible clinical indications” would be better “as close as possible to the clinical settings”

25. In line 613: “thoracotomy and blood treatment of the remaining blood in the CPB” would be better “thoracotomy and treatment of the remaining blood in the CPB”

26. In line 613-614: “(ii) a abdominal visceral model” should be “(ii) an abdominal visceral model”

27. In line 619: “blood treatment processing” should be “blood processing”

28. In line 620: “two cycle treatment” should be “two cycle treatments”

29. In line 622: “compatible with the surgery” would be better “compatible with the surgical time”

30. In line 625: “in human medicine” would be better “in the clinical practice”

31. In lines 636-637: “but was cleared over 90%” would be better “but over 90% could be cleared”

32. In lines 637-638: “This hemolysis phenomenon also explains the lesser performance of free hemoglobin removal compared to the heparin washout” would be better “Hemolysis may also explain the lower removal performance of hemoglobin compared to the heparin washout”

33. In lines 640-643: the sentence “Preliminary results with the present system…” is difficult to understand. Elaborate better. The sentence was eliminated as these data were extracted from the feasibility experiments which cannot be exploited in this study.

34. In line 644: “hemolysis in the treated blood was higher” should be “hemolysis in the treated blood is higher”

35. In line 645: “in vitro study on human blood testing the system” would be better “in vitro study testing the system with human blood”

36. In line 650: “compared to human blood treatment” should be “compared to human blood”

37. In line 651-652: “explained by blood shedding, coagulation activation during abdominal bleeding and suction conditions” should be “explained by the blood shedding, the coagulation activation during abdominal bleeding and the suction conditions”

38. In line 655-656: “Despite the hemolysis superior to the 0.8% threshold and RBCs yield below 80%” should be “Despite the hemolysis rate higher than the 0.8% threshold and the RBCs yield below 80%”

39. In lines 658-659: “free hemolysis” should be “free hemoglobin”

40. In line 663: “the surgery the concentration of red blood cells” should be “the surgery, to concentrate not only red blood cells”

41. In lines 667-670: “platelet yield always greater than 40% and greater than the one obtained in the preliminary i-Sep study on in vitro human blood” would be better “platelet yield was always greater than 40% and higher than that obtained in the preliminary in vitro study of i-Sep using human blood”

42. In line 678: “Some of the platelet yield over 100%” should be “Some of the platelet yields were over 100%”

43. In line 686: “whichever the animal model” would be better “in either model”

44. In lines 687-688: “any thrombus in any of them” would be better “any evidence of thrombus formation”

45. In line 691: “the incapacity of the treated blood to coagulate” would be better “the inability of the treated blood to coagulate”

46. In lines 699-700: “allowed not only RBCs concentration but also platelet one” would be better “allowed to concentrate not only RBCs but also platelets”

47. In line 701: “pro-coagulant effect nor cause” should be “pro-coagulant effect nor did cause”

48. In line 702: “cardiac or visceral” should be “cardiac or visceral bleeding”

6. PLOS authors have the option to publish the peer review history of their article (what does this mean?). If published, this will include your full peer review and any attached files.

Do you want your identity to be public for this peer review? For information about this choice, including consent withdrawal, please see our Privacy Policy.

Reviewer #1: No

Reviewer #2: No

Reviewer #3: No

 We have check that the figures meet the PLOS requirement using PACE diagnostic tool.

---

## [Decision Letter · Decision Letter 1]

26 Oct 2021

PONE-D-21-23852R1A novel autotransfusion device saving erythrocytes and platelets by filtration used in a 72 h survival swine model of controlled blood loss: perioperative hematologic and coagulation assessments, salvaged blood characteristics and system performancePLOS ONE

Dear Dr. Touzot-Jourde,

Thank you for submitting your manuscript to PLOS ONE. After careful consideration, we feel that it has merit but does not fully meet PLOS ONE’s publication criteria as it currently stands. Therefore, we invite you to submit a revised version of the manuscript that addresses the points raised during the review process. Please submit your revised manuscript by Dec 10 2021 11:59PM. If you will need more time than this to complete your revisions, please reply to this message or contact the journal office at plosone@plos.org. Please include the following items when submitting your revised manuscript:A rebuttal letter that responds to each point raised by the academic editor and reviewer(s). You should upload this letter as a separate file labeled 'Response to Reviewers'.A marked-up copy of your manuscript that highlights changes made to the original version. You should upload this as a separate file labeled 'Revised Manuscript with Track Changes'.An unmarked version of your revised paper without tracked changes. You should upload this as a separate file labeled 'Manuscript'.If applicable, we recommend that you deposit your laboratory protocols in protocols.io to enhance the reproducibility of your results. Protocols.io assigns your protocol its own identifier (DOI) so that it can be cited independently in the future. For instructions see: https://journals.plos.org/plosone/s/submission-guidelines#loc-laboratory-protocols. Additionally, PLOS ONE offers an option for publishing peer-reviewed Lab Protocol articles, which describe protocols hosted on protocols.io. Read more information on sharing protocols at https://plos.org/protocols?utm_medium=editorial-email&utm_source=authorletters&utm_campaign=protocols.

We look forward to receiving your revised manuscript.

Kind regards,

Ahmet Emre Eşkazan, M.D.

Academic Editor

PLOS ONE

Journal Requirements:

Reviewers' comments:

Reviewer's Responses to Questions

**Comments to the Author**

1. If the authors have adequately addressed your comments raised in a previous round of review and you feel that this manuscript is now acceptable for publication, you may indicate that here to bypass the “Comments to the Author” section, enter your conflict of interest statement in the “Confidential to Editor” section, and submit your "Accept" recommendation.

Reviewer #2: All comments have been addressed

Reviewer #3: All comments have been addressed

2. Is the manuscript technically sound, and do the data support the conclusions?

Reviewer #2: Yes

Reviewer #3: Yes

3. Has the statistical analysis been performed appropriately and rigorously? 

Reviewer #2: Yes

Reviewer #3: N/A

4. Have the authors made all data underlying the findings in their manuscript fully available?

Reviewer #2: Yes

Reviewer #3: Yes

5. Is the manuscript presented in an intelligible fashion and written in standard English?

Reviewer #2: Yes

Reviewer #3: Yes

6. Review Comments to the Author

Reviewer #2: (No Response)

Reviewer #3: I believe the paper was significantly improved with the revision.

There are however some points that need to be addressed, as follows.

Major points

The title is too long. Is it possible to shorten?

The Materials and Methods section is too long, and very exhaustive to read. It would benefit of shortening to about a half or 1/3, eventually by moving part to the supplemental material.

Compared to the visceral model, the platelet concentration was higher in the cardiac model, but as expected, the platelet yield was lower, which can be attributed to the deleterious effect of CBP on platelets, as discussed by the authors. The authors mention that there was no platelet activation during the treatment, and platelet kept the ability to be activated by treatment, confirmed in their previous study with human blood. Did the authors confirm this finding with the pig platelets? If yes, the results should be described, and if not, it should be mentioned in the Discussion.

In Page 30, Line 684, there is mention to the hemolysis rate higher than the 0.8% threshold. However, there is no mention to the expected hemolysis rate in the Materials and Methods section. It should be described in Page 17, together with the other expected device performance indexes.

Minor points

In Page 14, Lines 326-328: [“Use of International Standard ISO 10993-1”, “Biological evaluation of medical devices – Part 1: Evaluation and testing within a risk management process”] should be [Use of International Standard ISO 10993-1, “Biological evaluation of medical devices – Part 1: Evaluation and testing within a risk management process”]

In Page 19, Line 450: (Table S1) should be (Tables S1 and S2, respectively)

In Page 25, Lines 586 and 587: [61.7 % (50.1-71.9)] should be [61.7 (50.1-71.9) %] and [70.5 % (65.4-72.1)] should be [70.5 (65.4-72.1) %]

In Table 4: the hemoglobin concentration should be 80 and 61 (g/L) and not 8.0 and 6.1

In Page 27, Line 612: [complication or nor] should be [complication nor]

In Page 28, Line 625: [ovine RBCs cells] should be [ovine RBCs]

In Page 37, Line 857: is [Internet] necessary? Please check

In Page 37, Line 868: the link is incomplete. Please check

In Page 39, Line 902: the link is incomplete. Please check

In Page 39, Line 904: is [Internet] necessary? Please check

7. PLOS authors have the option to publish the peer review history of their article (what does this mean?). If published, this will include your full peer review and any attached files.

Reviewer #2: No

Reviewer #3: No

---

## [Author Response · Author response to Decision Letter 1]

2 Nov 2021

Please find below the responses to Reviewer 3.

Thank you very much for your comments and suggestions.

We have done our best to follow them.

Best regards

Reviewer #3: I believe the paper was significantly improved with the revision.

There are however some points that need to be addressed, as follows.

Major points

- The title is too long. Is it possible to shorten?

The title has been shortened and is now as follow : 

“A novel autotransfusion device saving erythrocytes and platelets used in a 72h survival swine model of surgically induced controlled blood loss”

- The Materials and Methods section is too long, and very exhaustive to read. It would benefit of shortening to about a half or 1/3, eventually by moving part to the supplemental material 

The Material and Methods section has been shortened. The details on the anesthesia and postoperative management have been transferred in the supplemental data section.

- Compared to the visceral model, the platelet concentration was higher in the cardiac model, but as expected, the platelet yield was lower, which can be attributed to the deleterious effect of CBP on platelets, as discussed by the authors. 

Thank you for the acknowledgement

- The authors mention that there was no platelet activation during the treatment, and platelet kept the ability to be activated by treatment, confirmed in their previous study with human blood. Did the authors confirm this finding with the pig platelets? If yes, the results should be described, and if not, it should be mentioned in the Discussion.

We could not test platelet activation in the transfusion bag after the treatment because we had no access to specific platelet activation evaluation test. Indeed, plasma protein are washed and eliminated in the treated blood. Then fibrinogen was eliminated and aggregation tests with the ROTEM platelets were not suitable. Flow cytometry could have been used as in the previous study on human blood. However, no flow cytometer was available on the experimentation site. The following sentence was added to the manuscript:

“The reinfusion of platelets is considered as a potential prothrombotic risk. However, it was shown that the i-SEP ATS does not activate platelets during the treatment, and that platelets keep their ability to be activated after the treatment [30]. However, platelet activation was not evaluated in this animal study.”

- In Page 30, Line 684, there is mention to the hemolysis rate higher than the 0.8% threshold. However, there is no mention to the expected hemolysis rate in the Materials and Methods section. It should be described in Page 17, together with the other expected device performance indexes.

The 0.8% refers to the maximal expected hemolysis in packed red cells defined by the European guidelines (European Directorate for the Quality of Medicines and HealthCare of the Council of Europe (EDQM): Guide to the Preparation, Use and Quality Assurance of Blood Components, 20th edition. Strasbourg, France, Council of Europe Publishing, 2020). The reference of the guidelines and the references are added to the manuscript as follow :

“Despite the hemolysis rate higher than the 0.8% threshold (as defined by the European guidelines, reference added) and the RBCs yield below 80 %, the treated blood reached the 45 % to 65 % hematocrit reference range and its reinfusion was well tolerated by the animals.”

Minor points

In Page 14, Lines 326-328: [“Use of International Standard ISO 10993-1”, “Biological evaluation of medical devices – Part 1: Evaluation and testing within a risk management process”] should be [Use of International Standard ISO 10993-1, “Biological evaluation of medical devices – Part 1: Evaluation and testing within a risk management process”]

In Page 19, Line 450: (Table S1) should be (Tables S1 and S2, respectively)

In Page 25, Lines 586 and 587: [61.7 % (50.1-71.9)] should be [61.7 (50.1-71.9) %] and [70.5 % (65.4-72.1)] should be [70.5 (65.4-72.1) %]

In Table 4: the hemoglobin concentration should be 80 and 61 (g/L) and not 8.0 and 6.1

In Page 27, Line 612: [complication or nor] should be [complication nor]

In Page 28, Line 625: [ovine RBCs cells] should be [ovine RBCs]

In Page 37, Line 857: is [Internet] necessary? Please check

In Page 37, Line 868: the link is incomplete. Please check

In Page 39, Line 902: the link is incomplete. Please check

In Page 39, Line 904: is [Internet] necessary? Please check

All the minor points have been addressed.

Concerning internet links, after a new careful reading of the guidelines, our interpretation is that the journal prefers having the full link. However it is a misreading on our part, we have identified the section to be remove in yellow in the document “revised manuscript”

---

## [Decision Letter · Decision Letter 2]

18 Nov 2021

A novel autotransfusion device saving erythrocytes and platelets used in a 72 h survival swine model of surgically induced controlled blood loss

PONE-D-21-23852R2

Dear Dr. Touzot-Jourde,

We’re pleased to inform you that your manuscript has been judged scientifically suitable for publication and will be formally accepted for publication once it meets all outstanding technical requirements.

There some inconsistencies regarding the references, and please carefully check the reference numbers to be listed sequentially throughout the paper during the proof editing. Also there are some typos, which needs to be checked and corrected.

Kind regards,

Ahmet Emre Eşkazan, M.D.

Academic Editor

PLOS ONE

Additional Editor Comments (optional):

Reviewers' comments:

Reviewer's Responses to Questions

**Comments to the Author**

1. If the authors have adequately addressed your comments raised in a previous round of review and you feel that this manuscript is now acceptable for publication, you may indicate that here to bypass the “Comments to the Author” section, enter your conflict of interest statement in the “Confidential to Editor” section, and submit your "Accept" recommendation.

Reviewer #3: All comments have been addressed

2. Is the manuscript technically sound, and do the data support the conclusions?

Reviewer #3: Yes

3. Has the statistical analysis been performed appropriately and rigorously? 

Reviewer #3: N/A

4. Have the authors made all data underlying the findings in their manuscript fully available?

Reviewer #3: Yes

5. Is the manuscript presented in an intelligible fashion and written in standard English?

Reviewer #3: Yes

6. Review Comments to the Author

Reviewer #3: I believe the comments were appropriately addressed, and now it is easier to read.

There are only small corrections that need to be done before acceptance, as follows:

Due to restructuring/shortening of the Materials and Methods section, the references need to be renumbered. It jumps from [30] in Line 130 to [42] in Line 167.

In Line 480 (Table 4), the median Hemoglobin concentration value was appropriately corrected, but the min – max values also need correction: [8.0] [6.1] should be [80] [61]

In Line 607, reference [30] is cited 3 times [30][30][30] → please check

7. PLOS authors have the option to publish the peer review history of their article (what does this mean?). If published, this will include your full peer review and any attached files.

Reviewer #3: No

---

## [Editor Report · Acceptance letter]

15 Mar 2022

PONE-D-21-23852R2 

A novel autotransfusion device saving erythrocytes and platelets used in a 72 h survival swine model of surgically induced controlled blood loss. 

Dear Dr. Touzot-Jourde:

I'm pleased to inform you that your manuscript has been deemed suitable for publication in PLOS ONE. Congratulations! Your manuscript is now with our production department. 

Kind regards, 

on behalf of

Dr. Ahmet Emre Eşkazan 

Academic Editor

PLOS ONE